# A spontaneous mitonuclear epistasis converging on Rieske Fe-S protein exacerbates complex III deficiency in mice

Janne Purhonen[1,2], Vladislav Grigorjev[1], Robert Ekiert [3], Noora Aho[4,5], Jayasimman Rajendran [1,2], Rafał Pietras [3], Katarina Truvé [6], Mårten Wikström [7], Vivek Sharma[4,7], Artur Osyczka[3], Vineta Fellman [1,2,8,9,10] & Jukka Kallijärvi [1,2,10]*

We previously observed an unexpected fivefold (35 vs. 200 days) difference in the survival of respiratory chain complex III (CIII) deficient $Bcs1l^{p.S78G}$ mice between two congenic backgrounds. Here, we identify a spontaneous homoplasmic mtDNA variant ($m.G14904A$, $mt$-$Cyb^{p.D254N}$), affecting the CIII subunit cytochrome b (MT-CYB), in the background with short survival. We utilize maternal inheritance of mtDNA to confirm this as the causative variant and show that it further decreases the low CIII activity in $Bcs1l^{p.S78G}$ tissues to below survival threshold by 35 days of age. Molecular dynamics simulations predict D254N to restrict the flexibility of MT-CYB $ef$ loop, potentially affecting RISP dynamics. In $Rhodobacter$ cytochrome $bc_1$ complex the equivalent substitution causes a kinetics defect with longer occupancy of RISP head domain towards the quinol oxidation site. These findings represent a unique case of spontaneous mitonuclear epistasis and highlight the role of mtDNA variation as modifier of mitochondrial disease phenotypes.

[1] Folkhälsan Institute of Genetics, Folkhälsan Research Center, P.O. Box 63 (Haartmaninkatu 8), FI-00014 Helsinki, Finland. [2] Stem Cells and Metabolism Research Program, Faculty of Medicine, University of Helsinki, Helsinki, Finland. [3] Department of Molecular Biophysics, Faculty of Biochemistry, Biophysics, and Biotechnology, Jagiellonian University, ul. Gronostajowa 7, 30-387 Kraków, Poland. [4] Department of Physics, University of Helsinki, P.O. Box 64 (Gustaf Hällströmin katu 2), FI-00014 Helsinki, Finland. [5] Department of Chemistry, University of Jyväskylä, P.O. Box 35 (Survontie 9B), FI-40014 Jyväskylä, Finland. [6] Sahlgrenska Academy, University of Gothenburg, P.O. Box 413 (Medicinaregatan 3), 41390 Gothenburg, Sweden. [7] Institute of Biotechnology, University of Helsinki, PL 56 (Viikinkaari 9), FI-00014 Helsinki, Finland. [8] Department of Clinical Sciences, Pediatrics, BMC F12, Lund University, 221 84 Lund, Sweden. [9] Children's Hospital, Helsinki University Hospital, P.O. Box 281 (Stenbäckinkatu 11), FI-00029 Helsinki, Finland. [10]These authors contributed equally: Vineta Fellman, Jukka Kallijärvi. *email: jukka.kallijarvi@helsinki.fi

Mitochondrial respiratory chain complex III (cytochrome $bc_1$ complex, CIII) oxidizes coenzyme Q, reduces cytochrome $c$, and translocates protons to generate membrane potential for ATP synthesis[1]. The mitochondrial inner membrane AAA-family translocase BCS1L, frequently mutated in CIII deficiency[2–4], is required for the topogenesis of the electron-transferring RISP (Rieske iron-sulphur protein, UQCRFS1) subunit and its assembly into CIII[5–7]. Homozygous $Bcs1l^{c.A232G}$ ($Bcs1l^{p.S78G}$) knock-in mice[8], carrying a GRACILE syndrome[9,10] patient mutation[3], recapitulate many manifestations of human CIII deficiency. They display post-weaning growth failure, hepatopathy, renal tubulopathy, and, in a C57BL/6JBomTac-derived genetic background (Lund colony), deterioration due to metabolic crisis with extreme hypoglycemia by 35 days of age (P35)[11–13]. When we brought the mutant strain to another facility via embryo transfer and bred it with the closely related C57BL/6JCrl background (Helsinki colony), the homozygotes developed similar early visceral and systemic manifestations but, unexpectedly, survived the early metabolic crisis and lived fivefold longer, to up to 200 days[14,15]. Suspecting genetic drift in the isolated Lund colony, we hypothesized that (a) homozygous genetic change(s) underlie the highly consistent survival difference between the two colonies. Here we perform whole-genome sequencing (WGS) to identify candidate variants, followed by a simple genetic experiment to show that a spontaneous mitochondrial DNA (mtDNA) variant underlies the survival difference. We combine mouse phenotyping, computational, and spectroscopic data to show how the effects of this non-pathogenic variant and the disease-causing $Bcs1l$ mutation converge to exacerbate CIII deficiency and disease progression.

## Results

**Short-lived $Bcs1l^{p.S78G}$ mice carry a novel mtDNA variant.** WGS ($n = 3$ for Lund C57BL/6JBomTac and $n = 2$ for Helsinki C57BL/6JCrl) revealed 844 homozygous single-nucleotide polymorphisms and 3655 small insertion/deletions between the strains, only 8 of which were in coding regions of genes (Supplementary Table 1). One of these was an mtDNA variant (m. G14904A) not present in any *Mus musculus* sequence in Gen-Bank. Genotyping of approximately 80 mice throughout past generations using archived genomic DNA from ear biopsies revealed that the variant was introduced from wild-type (WT) C57BL/6JBomTac females repeatedly after 2008, when congenization of the $Bcs1l^{c.A232G}$ knock-in allele was started (Supplementary Fig. 1a). Inspection of the pedigrees of the mt-Cyb-genotyped mice showed that two early-generation females had given birth to both WT and variant-carrying progeny, suggesting initial heteroplasmy. However, sequencing of 346 bacterial clones of *mt-Cyb* PCR amplicon from the liver, kidney, heart, and skeletal muscle DNA showed no sign of heteroplasmy in somatic tissues in later generations (Supplementary Fig. 1b). The fact that the variant was homoplasmic in an apparently healthy WT mouse colony suggested that it must be non-pathogenic. Indeed, analysis of mtDNA sequences deposited to GenBank showed that the three-toed sloth species (*Bradypus*) of South America, known for their very low metabolic rate[16,17] naturally carry this variant (Fig. 1a). Intriguingly, the variant affects the mtDNA-encoded CIII subunit cytochrome b, changing a conserved aspartate 254 to asparagine ($mt$-$Cyb^{p.D254N}$). A negatively charged amino acid in this position is highly conserved across eukaryotes and aerobic prokaryotes (Fig. 1a), with limited conservancy in archaea. Therefore, as $mt$-$Cyb^{p.D254N}$ potentially directly affects CIII function, it appeared as a most likely genetic modifier of the survival of $Bcs1l^{p.S78G}$ mice.

**$mt$-$Cyb^{p.D254N}$ dictates the short survival of $Bcs1l^{p.S78G}$ mice.** We utilized maternal inheritance of mtDNA and crossbred $Bcs1l^{p.S78G}$ heterozygotes from the two inbred colonies to obtain F1 progeny carrying otherwise equalized nDNA background (heterozygous for all initially homozygous alleles differing between the colonies) and either WT or variant mitochondria (Fig. 1b). The homozygous $Bcs1l^{p.S78G}$ progeny of females from the Lund colony (C57BL/6JBomTac) had median survival of 38 days while that of females from the Helsinki colony (C57BL/6JCrl) 150 days (Fig. 1c). As WGS showed no other mtDNA differences between the strains, the crossbreeding unequivocally confirmed that the maternally inherited $mt$-$Cyb^{p.D254N}$ variant was the major determinant of the survival of $Bcs1l^{p.S78G}$ mice.

We have previously described the early-onset disease manifestations (growth restriction, hypoglycemic, hepatopathy, and kidney tubulopathy) of $Bcs1l^{p.S78G}$ mice in both the short- and long-living strains separately[8,13–15,18,19]. Here we reassessed some of the main manifestations in the genetically comparable F1 mice. The presence of $mt$-$Cyb^{p.D254N}$ aggravated the growth restriction of the mutant mice (Fig. 1d). Remarkably, $mt$-$Cyb^{p.D254N}$ alone also slightly decreased the weight of the F1 juvenile $Bcs1l$ WT progeny. All $Bcs1l^{p.S78G}$ mice had abnormally low blood glucose, but in those carrying $mt$-$Cyb^{p.D254N}$ the blood glucose was below quantification limit in several individuals, indicating severe hypoglycemia expected to lead to coma and death (Fig. 1e). Histopathological analysis revealed hepatopathy with glycogen depletion and incipient fibrosis that was similar in both mutant genotypes up to age P29–P33 (Supplementary Fig. 2). However, in the very end-stage (>P34) livers, diffuse hepatocyte death was present, indicating rapid deterioration to fulminant hepatic failure in the $mt$-$Cyb^{p.D254N}$-carrying mutants (Supplementary Fig. 3). Staining for cleaved caspase 3, a standard marker of apoptosis, confirmed massive hepatic cell death in the end-stage $mt$-$Cyb^{p.D254N}$-carrying mutants (Fig. 1g, h, Supplementary Fig. 3). This time point coincided with near-complete exhaustion of hepatic ATP (Fig. 1f). In contrast, only few dying hepatocytes were present in comparable liver sections from $Bcs1l^{p.S78G}$ mice with WT mitochondria (Fig. 1g, h). Instead of rapid ATP depletion, these mice showed a trend toward normalization after P34 (Fig. 1f). These findings are in line with our previous studies[14,15] showing that the hepatopathy of $Bcs1l^{p.S7Fig.8G}$ mice in the long-living colony does not progress to liver failure even at end stage (P150–P200).

The kidneys of all $Bcs1l^{p.S78G}$ mice were atrophic but luminal casts suggesting proteinuria were obvious only when these mice carried $mt$-$Cyb^{p.D254N}$, indicating more advanced kidney disease (Supplementary Fig. 4).

**$mt$-$Cyb^{p.D254N}$ exacerbates CIII deficiency in $Bcs1l^{p.S78G}$ mice.** After the crossbreeding experiment showed unequivocally that $mt$-$Cyb^{p.D254N}$ was the major determinant of the survival of $Bcs1l^{p.S78G}$ mice, we set out to investigate its effect on CIII activity. We have previously shown[8] that, in the short-living background, the progressive CIII dysfunction is severest in the liver, the activity being normal at birth but thereafter rapidly decreasing to as low as 10% of WT activity near end stage (P33–P35). The first symptoms appear soon (P24) after weaning at approximately 50% residual activity. Below 25% residual activity, the mice become severely hypoglycemic and die within a few days[8,13]. We measured CIII activity in the key affected (liver, kidney) and histologically non-affected (skeletal muscle, heart) tissues in the F1 mice and found that the low CIII activity was further decreased by $mt$-$Cyb^{p.D254N}$ in all four tissues (Fig. 2). The mean hepatic CIII activity at P33 was 9% of WT values in $mt$-$Cyb^{p.D254N}$;$Bcs1l^{p.S78G}$ mice, well below the survival threshold

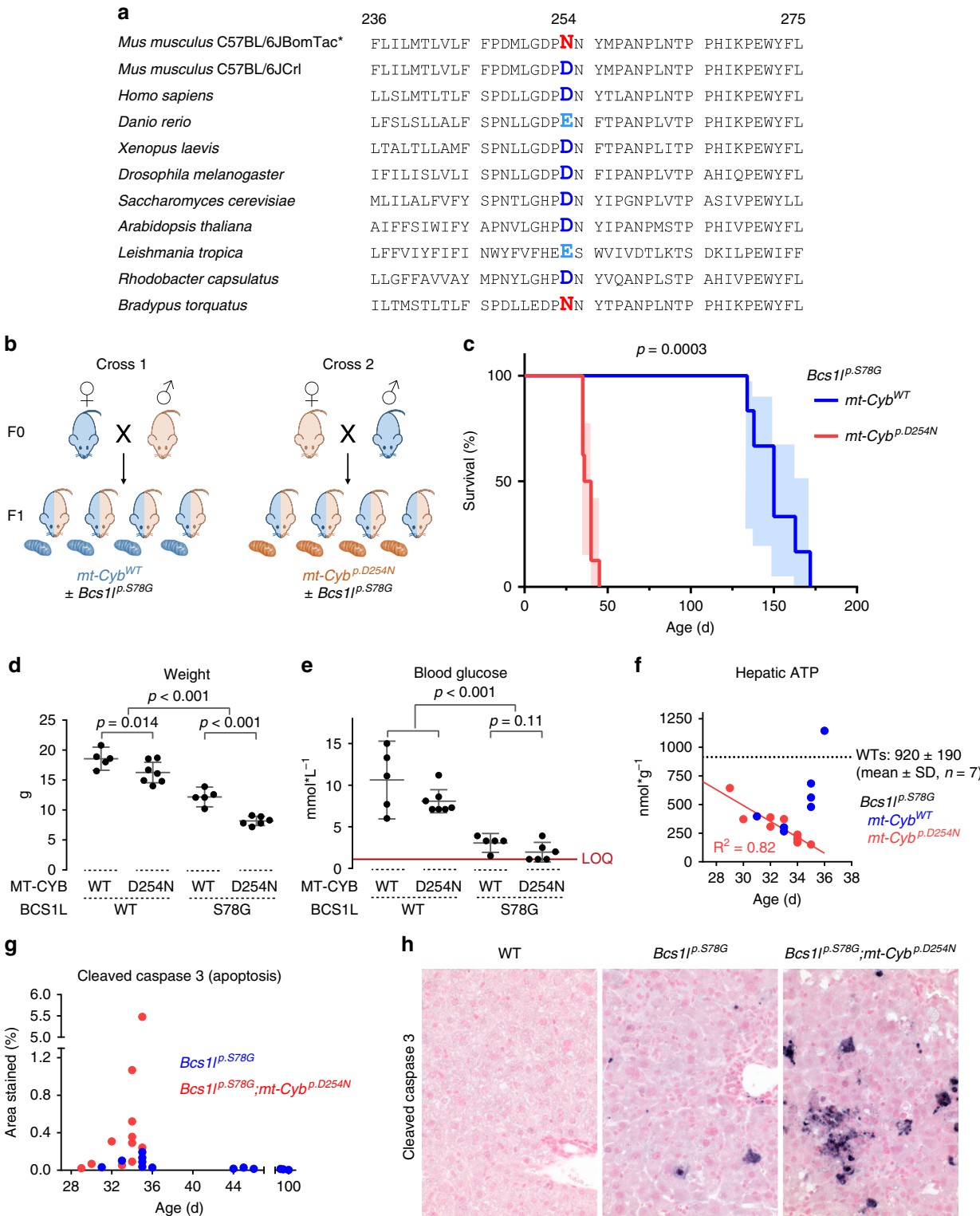

**Fig. 1 Identification of a spontaneous mtDNA variant as a modifier of *Bcs1l* mutant phenotype. a** Alignment of amino acids 236–275 (numbering for *Mus musculus* sequence) of MT-CYB protein of various organisms. Asterisk (*), Lund in-house C57BL/6JBomTac-derived mouse colony. **b** Cartoon of the crosses to assess the effect of the maternally inherited mtDNA on the survival of *Bcs1l*$^{p.S78G}$ mice. **c** Survival analysis of *Bcs1l*$^{p.S78G}$ mice with either the C57BL/6JBomTac* ($n = 8$) or C57BL/6JCrl (Helsinki colony) maternal parent ($n = 6$). Light blue- and red-shaded areas illustrate 95% CI. **d** Weight and **e** blood glucose of juvenile F1 males. Three *mt-Cyb*$^{p.D254N}$-carrying *Bcs1l*$^{p.S78G}$ mice had blood glucose below the limit of quantification (LOQ, 1 mmol/l) of the glucose meter. **f** Hepatic ATP concentration of *Bcs1l* mutant mice with and without the *mt-Cytb*$^{p.D254N}$ variant plotted against mouse age. **g** The degree of hepatic apoptosis, cleaved caspase-3-positive cross-sectional area, plotted against age. **h** Representative micrographs of immunostained liver sections for cleaved caspase 3 from 34- to 35-day-old mice. Scale bar represents 100 μm. Statistics: **c** log-rank (Mantel–Cox) test, **d** one-way ANOVA followed by planned comparisons, **e** Kruskal–Wallis followed by Mann–Whitney *U* tests. Error bars represent 95% CI of the mean in all figures.

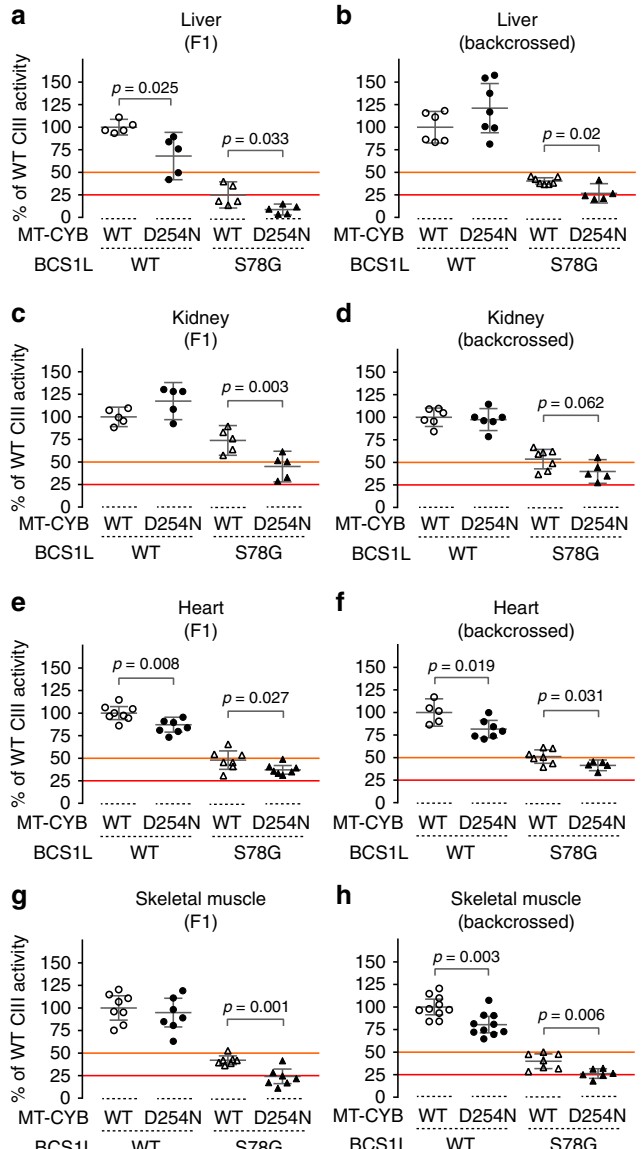

**Fig. 2 Synergistic effect of *Bcs1l*[p.S78G] and *mt-Cyb*[p.D254N] on CIII activity. a–h** CIII activity in liver (**a**, **b**) and kidney mitochondrial fractions (**c**, **d**) and heart (**e**, **f**) and skeletal muscle homogenates (**g**) from F1 hybrid progeny (**a**, **c**, **e**, **g**) and from mice backcrossed to C57BL/6JCrl (**b**, **d**, **f**, **h**). CIII activity data from isolated liver and kidney mitochondria were normalized to mitochondrial protein amount. The data from muscle and heart homogenates were normalized to relative respiratory chain content as assessed by measuring cyanide-sensitive cytochrome c oxidation. Cardiac and skeletal muscle homogenate CIII activity relative to tissue protein are shown in Supplementary Fig. 5. The orange line represents 50% residual activity, a threshold for appearance of hypoglycemic and hepatic manifestations in *Bcs1l* mutant mice. The red line represents 25% residual activity, a value observed in the liver of *Bcs1l* mutant mice in the colony with short survival at the onset of the terminal stage. Statistics: one-way ANOVA followed by planned comparisons. Error bars represent 95% CI of the mean.

(Fig. 2a). The mean hepatic CIII activity remained at 25% of WT in *Bcs1l*[p.S78G] homozygotes with WT *mt-Cyb* (Fig. 2a, c), as also shown in our earlier studies. Furthermore, the mean CIII activity of *Bcs1l*[p.S78G] homozygotes with WT mtDNA did not decrease <50% in the kidney and heart, whereas in *mt-Cyb*[p.D254N] carriers

it did (Fig. 2c, f). Further measurements from a panel of mice backcrossed several times to the C57BL/6JCrl background confirmed the synergistic effect of *Bcs1l*[p.S78G] and *mt-Cyb*[p.D254N] on CIII activity (Fig. 2c, d, g, h and Supplementary Fig. 5). Interestingly, *mt-Cyb*[p.D254N] alone consistently decreased cardiac CIII activity in both sample panels (Fig. 2f, g, Supplementary Fig 5), whereas this trend was not as obvious in other tissues (Fig. 2). For comparison, we analyzed the enzymatic activities of other respiratory chain complexes in isolated liver and kidney mitochondria from the F1 mice (Supplementary Fig. 6). CI and CIV activities were similar in all groups. CII activity was slightly lower in the *Bcs1l*[p.S78G];*mt-Cyb*[p.D254N] liver as compared to the other genotypes, while in the kidney, CII activity was increased by sole *mt-Cyb*[p.D254N] or *Bcs1l*[p.S78G] but not by their combination. In sum, *mt-Cyb*[p.D254N] further compromised CIII activity in *Bcs1l*[p.S78G] mice, decreasing it below survival threshold.

Aspartate 254 is located in the MT-CYB *ef* loop involved in RISP binding[20,21], which raised the question whether its substitution to asparagine might exacerbate the RISP assembly defect caused by the *Bcs1l* mutation. To quantify RISP assembled into CIII, we subjected digitonin-solubilized liver (Fig. 3a) and kidney (Fig. 3b) mitochondrial fractions to blue native gel electrophoresis. The steady-state level of RISP in CIII dimer and supercomplexes was markedly affected by the *Bcs1l* genotype, but with no consistent further impairment by *mt-Cyb*[p.D254N] in these tissues.

**mt-Cyb[p.D254N] modifies mitochondrial bioenergetics.** We have previously shown that CI&CII-linked oxidative phosphorylation (OXHPOS) is compromised in *Bcs1l*[p.S78G] mice in both the short- and long-living colonies[8,15,18]. To assess how *mt-Cyb*[p.D254N] might affect respiratory electron transfer and OXPHOS, we subjected liver and kidney mitochondria from the F1 hybrid mice (Supplementary Fig. 7) and a panel of mice backcrossed to C57BL/6JCrl (Fig. 4, Supplementary Figs. 8 and 9) to high-resolution respirometry. The CIII defect of *Bcs1l* mutant mice with or without *mt-Cyb*[p.D254N] did not compromise OXPHOS driven by sole NADH-producing substrates (CI-linked respiration) (Supplementary Figs. 7 and 8) as we have previously reported[8,15,18] and in line with known high threshold for CIII inhibition[22]. Therefore, we proceeded to assess maximal OXPHOS capacity via convergent electron flow to the coenzyme Q pool via CI and CII by subsequent addition of succinate (CI&CII-linked OXPHOS). This parameter strongly correlated with residual CIII activity (Supplementary Fig. 10) and revealed the CIII dysfunction in mice carrying *Bcs1l*[p.S78G] and a further exacerbation by *mt-Cyb*[p.D254N] (Fig. 4a and Supplementary Figs. 7 and 9).

After measuring CI&CII-linked phosphorylating respiration, we inhibited ATP synthase with oligomycin to induce leak respiration, a high membrane potential state in which limited respiration occurs due to proton leakage. Similar to CI&CII-linked phosphorylating respiration, the CI&CII-linked leak respiration was also low in *Bcs1l*[p.S78G] and even lower in *Bcs1l*[p.S78G];*mt-Cyb*[p.D254N] tissues (Fig. 4b and Supplementary Fig. 7). Interestingly, *mt-Cyb*[p.D254N] alone decreased CI&CII-linked leak respiration in kidney mitochondria (Fig. 4b and Supplementary Fig. 7b). We did not observe this in liver mitochondria from the F1 hybrids (Supplementary Fig. 7a), but in liver mitochondria from mice backcrossed to C57BL/6JCrl (Fig. 4b) the trend was significant and similar to that in the kidney.

In some tissues, the phosphorylating system (adenine nucleotide translocase, phosphate transporters, and the ATP synthase) limits the electron-transferring proton translocases and subsequent respiration. Therefore, we also measured maximal capacity of the electron transfer by dissipating the membrane potential

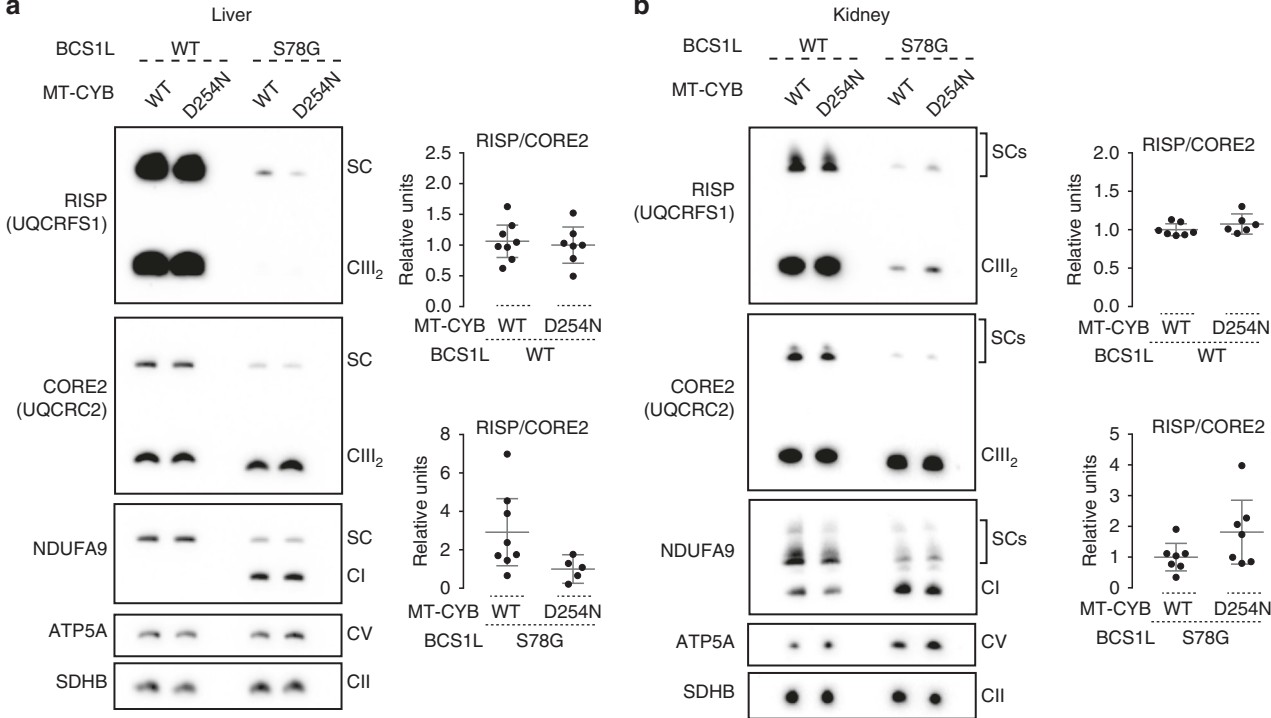

**Fig. 3 Analysis of RISP steady state level in CIII. a, b** BNGE analyses of digitonin-solubilized mitochondria from **a** liver and **b** kidney. ATP5A and SDHB are shown as loading controls. The individually analyzed and quantified samples were pooled together to prepare the representative blots. SC supercomplex. Error bars represent 95% CI of the mean.

with the protonophore carbonyl cyanide 4-(trifluoromethoxy) phenylhydrazone (FCCP). The rate of uncoupled CI&CII-linked respiration revealed that the respiration in *Bcs1l* mutant mitochondria with or without *mt-Cyb*[p.D254N] was limited by both the CIII defect and by the phosphorylating system (Fig. 4c and Supplementary Fig. 7). *mt-Cyb*[p.D254N] alone did not limit CI&CII-linked maximal electron transfer capacity. However, as the leak respiration was decreased, the *mt-Cyb*[p.D254N] mice showed elevated electron transfer coupling efficiency (Fig. 4d, e, Supplementary Fig 7), which is a ratio calculated from leak respiration and maximal electron transfer capacity. The decreased leak respiration and increased electron transfer coupling efficiency suggest that *mt-Cyb*[p.D254N] decreases proton leakage at CIII, or indirectly elsewhere, or it affects a rate-limiting step in CIII enzymatic cycle when it is working against high membrane potential.

Finally, as a measure of CII and CIII-derived reactive oxygen species production, we measured $H_2O_2$ emission under succinate-driven phosphorylating respiration with reverse electron flow to CI blocked by rotenone (Supplementary Fig. 11). Total $H_2O_2$ emission (per mitochondrial protein) was not significantly different between the groups. However, relative to oxygen consumption it was increased in the *Bcs1l*[p.S78G] mitochondria from the liver (median: 1.5% vs 0.8%, Mann–Whitney *U* test <0.0001) and kidney (0.8% vs 0.5%, *p* = 0.003) as compared to *Bcs1l* WTs.

**_mt-Cyb_[p.D254N] restricts RISP head domain movement.** As *mt-Cyb*[p.D254N] did not seem to have a consistent effect on CIII assembly, which is compromised by the *Bcs1l* mutation, a more subtle mechanism must underlie their synergistic effect on CIII activity in mice. We inspected published crystal structures of eukaryotic CIII to model structural changes that the D254N amino acid change might cause in MT-CYB or in the complex. These showed that the RISP head domain (RISP-HD) brushes the

*ef* loop segment of MT-CYB and docks to both the MT-CYB and CYTC1 subunits in two different conformations to promote electron transfer (Fig. 5a–c). Thus an amino acid substitution in the *ef* loop segment may lead to compromised RISP dynamics[23,24]. In order to obtain further microscopic insight, we performed fully atomistic classical molecular dynamics (MD) simulations of cytochrome $bc_1$ in membrane-solvent environment. These showed that D254N likely restricts the conformational flexibility in the surrounding region (Fig. 5d–g), displayed as the root mean square fluctuation of the segment T250 to V270 in the *ef* loop of MT-CYB (Fig. 5d), which strongly interacts with RISP. Accordingly, we observed a simultaneous reduction in mobility for the segment P136-W146 of RISP (Fig. 5e). A neutral asparagine in this position interacted with fewer water molecules in comparison to a charged aspartate (Fig. 5f) and displayed a lower side chain mobility (Fig. 5g). These data suggested that compromised mobility of the *ef* loop and the RISP-HD might affect a rate-limiting step in the catalytic cycle of CIII in *Bcs1l*[p.S78G] mutants in which the CIII assembly defect may lead to heterodimers[25] containing only one RISP molecule per CIII dimer. When simulations were performed by modeling a ubiquinol ($QH_2$) and ubiquinone (Q) at the $Q_o$ and $Q_i$ sites, respectively, we found that the flexibility of the domains around D254 locus was reduced in comparison to simulations without Q molecules (Supplementary Fig. 12). This is in part due to the brushing of *ef* loop residues with the $QH_2$ head group. In WT simulations, residues P270 and Y298 contacted the $QH_2$ head group (≤3.5 Å criteria) for ca. 70% and 90% of the simulation time, respectively. Similarly, in mutant simulations, P270 and Y298 interacted with $QH_2$, for 100 and 18% of total simulation time, with an additional interaction observed between $QH_2$ and E271 (48%). These interactions are likely to reduce the *ef* loop dynamics in tight coupling with substrate binding and dynamics. Nevertheless, a subtle difference in mobility was observed for the *ef* loop segment in agreement with the current experimental and earlier biochemical data[26].

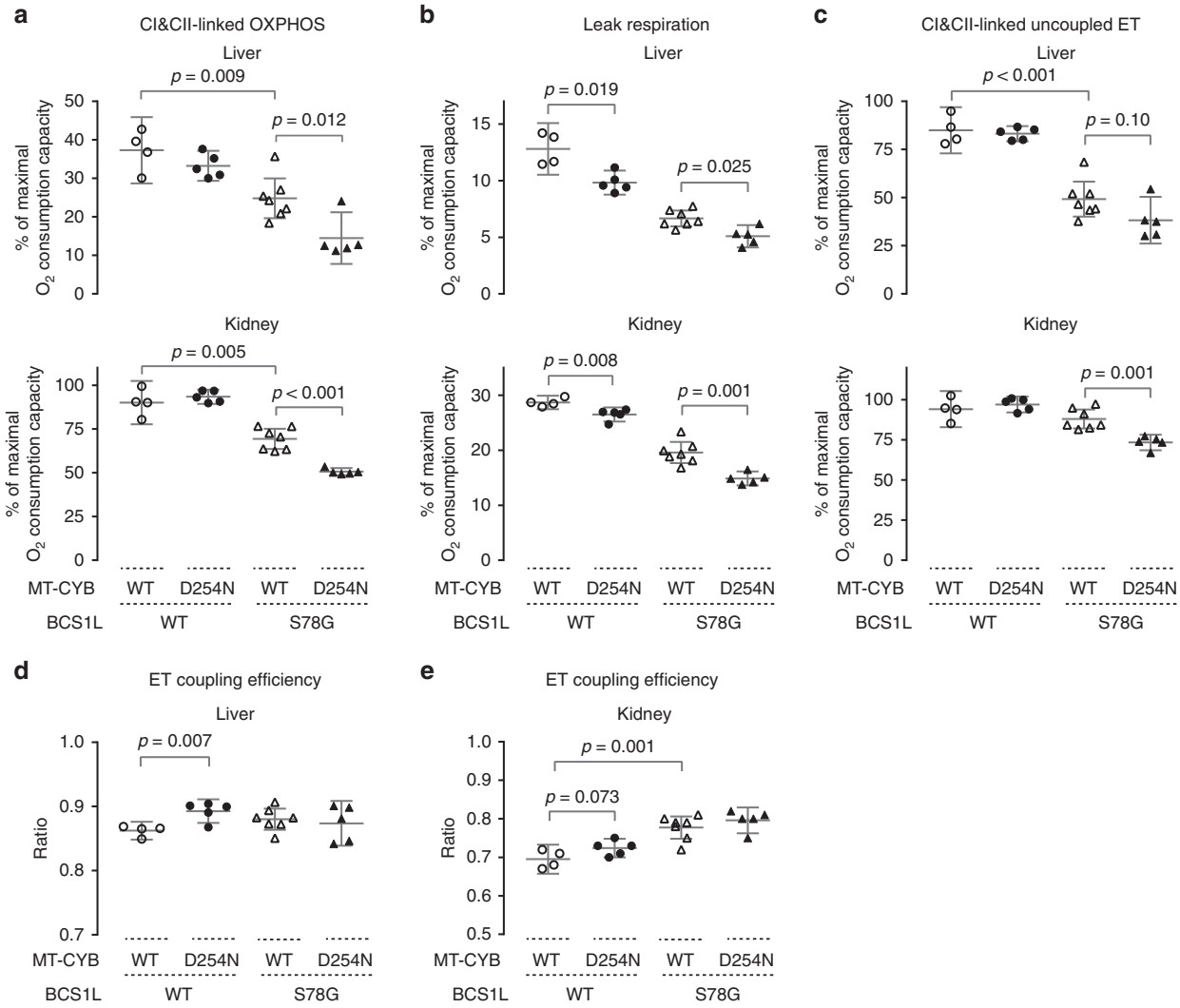

**Fig. 4 Effect of** $mt$-$Cytb^{p.D254N}$ **on oxidative phosphorylation in** $Bcs1l^{p.S78G}$ **and wild-type mice. a** Phosphorylating respiration of liver and kidney mitochondria upon convergent electron flow to coenzyme Q pool via CI and CII in presence of ADP (CI&CII-linked OXPHOS). Mitochondrial NADH was generated with a substrate cocktail comprising malate, pyruvate, and glutamate. Succinate served as direct substrate for CII. **b** Proton leak upon high membrane potential state was assessed after inhibition of ATP synthase by oligomycin. **c** Subsequently, a protonophore, FCCP, was titrated to assess the maximal uncoupled electron transfer (ET) capacity. Maximal oxygen consumption capacity, as measured using ascorbate and TMPD as substrates for the terminal oxidase (CIV), was set as the reference state for the oxygen fluxes (**a–c**). Supplementary Fig. 9 shows the same data relative to mitochondrial protein. **d**, **e** ET coupling efficiency (1−$L$/$E$) as calculated from oligomycin-induced leak respiration ($L$) and uncoupler-induced maximal CI&CII-linked electron transfer capacity ($E$). Samples collected at P30 from backcrossed mice were used for the analyses. Supplementary Fig. 7 shows similar analysis from F1 hybrid mice. Statistics: One-way ANOVA followed by planned comparisons. Error bars represent 95% CI of the mean.

To test the predictions of the in silico simulations, we employed the purple photosynthetic bacterium *Rhodobacter capsulatus*, in which the cytochrome b gene can be mutagenized and the simple three-component $bc_1$ complex purified and subjected to sensitive structural and functional measurements. To this end, we introduced the equivalent mutation, D278N, into *Rhodobacter* cytochrome b. As expected from the non-pathogenic nature of the variant in mice, the subunit composition, spectral properties, activity, and superoxide production of the mutant complex were not affected, and it sustained normal $bc_1$-dependent photosynthetic growth (Supplementary Fig. 13, Supplementary Table 2). To measure RISP movement, predicted by the MD simulations to be affected by the stiffened mutant *ef* loop, we monitored spectroscopically the enhancement of phase relaxation of RISP [2Fe-2S] cluster by oxidized heme $b_L$[27]. This method allows monitoring of changes in the distribution of RISP-HD positions between the $Q_o$ position and cytochrome $c_1$

position. The enhancement is strongest when the motion of RISP-HD is restricted predominantly toward positions at the $Q_o$ site, such as in the +2Ala mutant (Fig. 5h, i) that was used as one reference point. Unrestricted motion in the native enzyme resulted in a much broader distribution of RISP-HD positions and a relatively weak enhancement (Fig. 5h, i), providing us with another reference point. The D278N mutant displayed an increased enhancement compared to native enzyme but not as large as in the +2Ala. This indicates that the D278N mutation caused a subtle but remarkable effect on the motion, shifting the average position of RISP-HD toward the $Q_o$ position and confirming the predictions from the MD simulations of the eukaryotic complex.

**$mt$-$Cyb^{p.D254N}$ modifies whole-body metabolism.** To assess the metabolic effect of $mt$-$Cyb^{p.D254N}$ at whole-body level, we

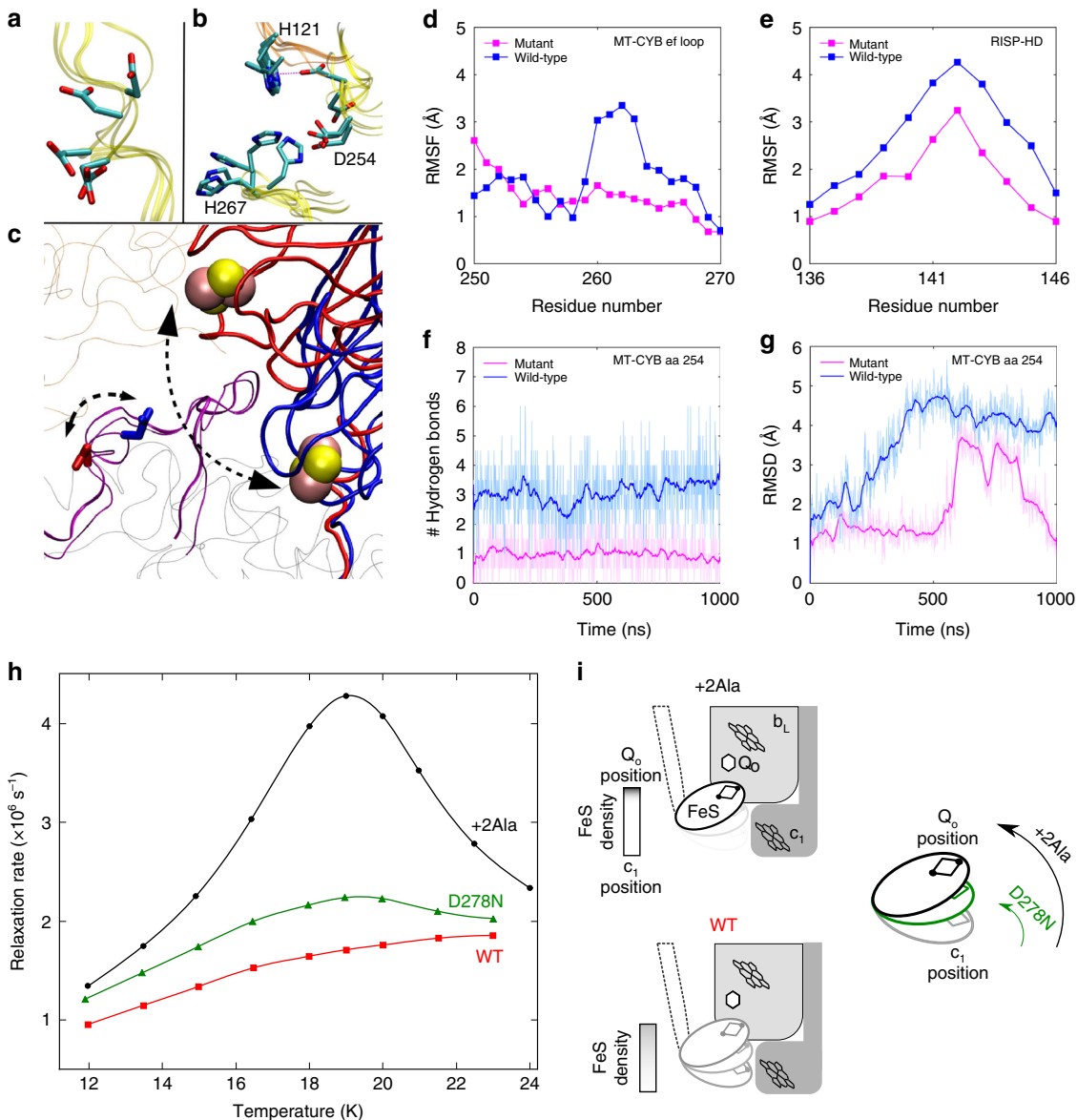

**Fig. 5 Effect of D254N of the flexibility of MT-CYB ef loop and RISP head domain movement. a** Conformational changes in the side chain of D254 of MT-CYB subunit (yellow ribbon) as observed in different crystal structures of the cytochrome $bc_1$ complex. **b** The side chain fluctuations occur with simultaneous changes in the surrounding residues of cytochrome $c_1$ (H121) and MT-CYB (H267). **c** Two interchangeable conformations of the Fe-S domain and D254, as seen in crystal structures 1BE3 (red) and 2FYU (blue), are marked by dotted arrows. The *ef* loop segment (residues 250–270) is shown in purple, and [2Fe-2S] clusters as pink-yellow spheres. **d** RMSF of the $C\alpha$ atoms from segment T250-V270 of the MT-CYB subunit. **e** RMSF of segment P136-W146 in the RISP subunit. **f** Number of hydrogen bonds between the side chain of D254/N254 (D255 in *S. cerevisiae* complex) and water molecules, with a criteria of a 3.0 Å donor–acceptor (D...A) distance and >160° bond angle (D-H....A). **g** RMSD of the D254/N254 side chain. **h** Temperature dependence of phase relaxation rate of the [2Fe–2S] cluster in native and mutant-purified *Rhodobacter* cytochrome $bc_1$. **i** Schematic presentation illustrating changes in equilibrium distribution of positions of RISP-HD. The native cytochrome $bc_1$ and +2Ala cytochrome b mutant served as references to which the enhancement of phase relaxation in D278N (corresponding to *Mus musculus* D254N) was compared. The +2Ala cytochrome *b* almost completely arrests RISP-HD at the $Q_o$ site, which results in strongest enhancement (black in **h**); native cytochrome $bc_1$ shows broader distribution of positions of RISP-HD and the enhancement is much weaker (red in **h**). Enhancement in D278N (green in **h**) indicates a shift of RISP-HD toward the $Q_o$ site (green arrow) compared to native enzyme, which however is significantly smaller than that of +2Ala (black arrow).

subjected the mice to indirect calorimetry (Fig. 6). *Bcs1l*^*p.S78G* homozygotes with WT mitochondria did not significantly differ from WT mice during daytime measurements, whereas *Bcs1l*^*p.S78G* homozygotes carrying *mt-Cyb*^*p.D254N* variant clearly did, as indicated by low respiratory exchange ratio (RER) and energy expenditure (Fig. 6a–c). Both parameters suggested a profound starvation-like metabolic state in these mice. The decreased energy expenditure remained highly significant also after treating body weight as a covariate. Mice are nocturnal animals and nighttime

whole-body metabolism data showed an even more distinct separation of *Bcs1l*^*p.S78G*;*mt-Cyb*^*p.D254N* mice from all other genotypes (Fig. 6a–c). Measurements during nighttime distinguished also *Bcs1l*^*p.S78G* mice with WT mitochondria and, surprisingly, those carrying *mt-Cyb*^*p.D254N* alone from WT mice. In fact, *mt-Cyb*^*p.D254N* variant and the *Bcs1l*^*p.S78G* mutation caused a similar decrease in RER and energy expenditure. Differences in physical activity did not explain these differences (Supplementary Fig. 14).

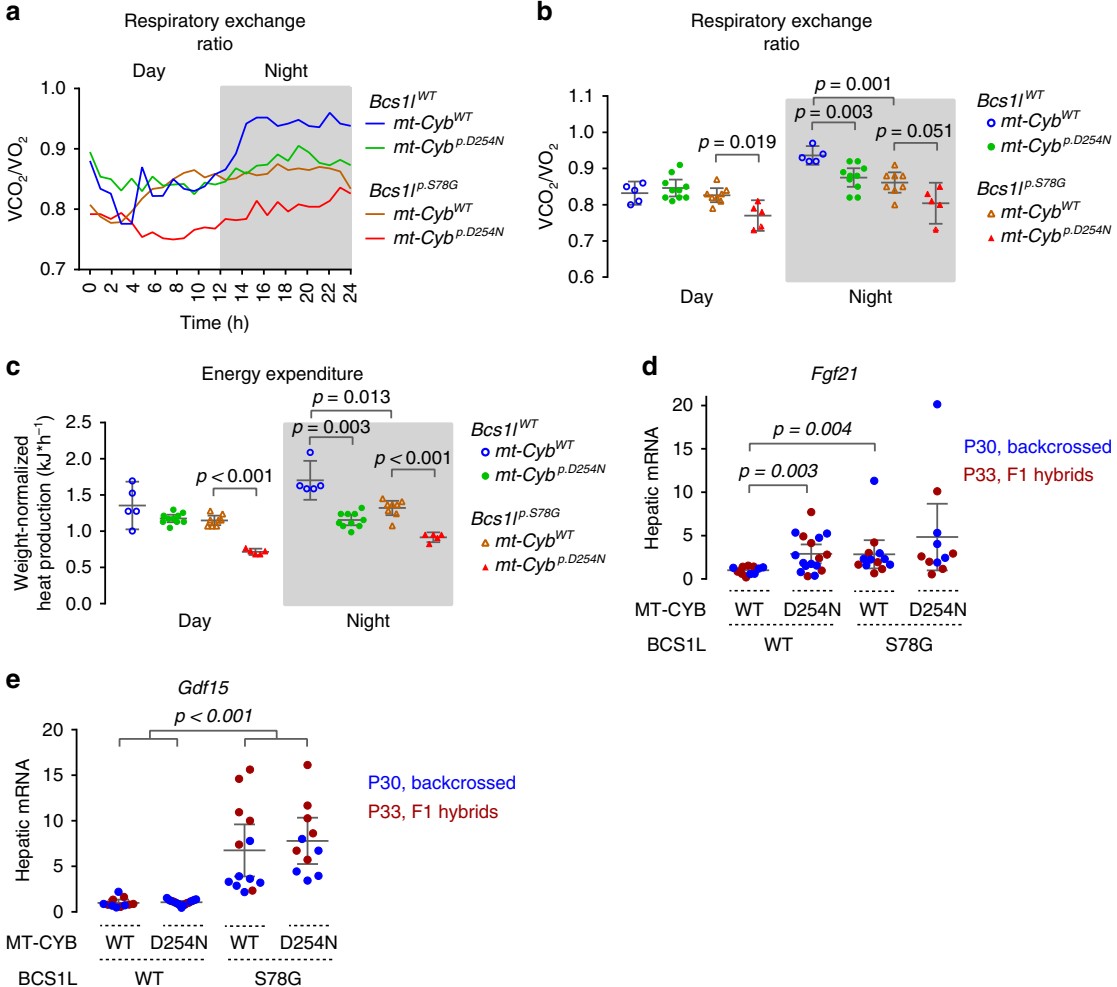

**Fig. 6 Whole-body metabolism in wild-type and *Bcs1l*^p.S78G mice with and without *mt-Cytb*^p.D254N.** Comprehensive Laboratory Animal Monitoring System (CLAMS) apparatus was used to measure $O_2$ consumption and $CO_2$ production by the four different mouse groups (backcrossed to C57BL/6JCrl) at age P28–29. **a** Circadian rhythm of respiratory exchange ratio. **b** Average 12-h respiratory exchange ratio of daytime and nighttime measurements. **c** Average 12-h energy expenditure estimated by indirect calorimetry for daytime and nighttime. The plotted values represent weight-normalized energy expenditure from ANCOVA model with genotype as a fixed factor and weight as a covariate. **d** Hepatic gene expression of a hepatocyte-derived mitochondrial stress-inducible endocrine factor, *Fgf21*. **e** Hepatic gene expression of *Gdf15*, a macrophage-secreted marker of mitochondrial dysfunction. Statistics: one-way ANOVA (**b**, **d**, **e**) or ANCOVA (**c**) followed by planned comparisons with Welch's *t* statistics.

To shed further light on the metabolic effects of *mt-Cyb*^p.D254N, we measured hepatic gene expression of *Fgf21*, a mainly hepatocyte-derived endocrine factor linked to mitochondrial stress and stressed energy metabolism[28], and *Gdf15*, another circulating marker of mitochondrial dysfunction[29]. We have previously shown that both are induced in *Bcs1l* mutant mice at later age[14,15]. Also here *Fgf21* and *Gdf15* expression were upregulated by the *Bcs1l* mutation (Fig. 6d, e), in line with expected metabolic stress and energy deprivation. Strikingly, *mt-Cyb*^p.D254N alone induced the expression of *Fgf21* to similar level as the pathogenic *Bcs1l* mutation (Fig. 6d), showing together with the respirometric and indirect calorimetric data that *mt-Cyb*^p.D254N causes distinct metabolic alterations despite being an apparently non-pathogenic variant.

## Discussion

We discovered an extreme genetic background effect in the survival of CIII deficient mice and tracked the cause to a spontaneous variant in the mtDNA-encoded CIII subunit MT-CYB. This overtly silent mtDNA variant likely arose >10 years ago in an isolated colony of C57BL/6JBomTac mice and became

homoplasmic by random drift in the germline, and finally expanded to the whole colony from a single common matrilinear ancestor. Incidentally, *Bcs1l* mutant mice with defective assembly of RISP subunit into CIII were generated and congenized in this facility[8]. This resulted in transfer of the mtDNA variant-carrying mitochondria into the *Bcs1l*^p.S78G mice and a hidden genetic interaction, exacerbating their CIII deficiency and causing a lethal metabolic crisis by 5–6 weeks of age. We discovered this only after transferring the mice to another facility and crossing them to a commercial C57BL/6JCrl strain, in which the homozygotes lived 4–5 times longer. We identified a candidate mtDNA variant, *mt-Cyb*^p.D254N, using WGS and utilized maternal inheritance of mtDNA to explicitly show that this variant was the main determinant of the survival.

In addition to *BCS1L* mutations, also *MT-CYB* mutations are a rare cause of CIII deficiency in humans. For example, mutations affecting the *ef* loop and $Q_o$ site cause CIII deficiency manifesting as cardiomyopathy and exercise intolerance[30]. A heteroplasmic in-frame deletion of 8 amino acids spanning D254, the amino acid affected in our mice, was reported in a patient with exercise intolerance, a relatively benign phenotype[31]. Aspartate 254 is located in

the RISP binding *ef* loop of MT-CYB, and patient and artificial experimental mutations in this region are known to affect either quinol oxidation and subsequent electron transfer or RISP assembly[23,24]. In our mice, *mt-Cyb*[p.D254N] did not consistently affect the steady-state level of RISP in CIII. Instead, MD simulations predicted that a neutral asparagine in this position stiffens the *ef* loop and restricts RISP-HD motion between the $Q_o$ and cytochrome $c_1$ sites. Pulse electron paramagnetic resonance (EPR) measurements in isolated *Rhodobacter* $bc_1$ complex confirmed this. In another bacterial system, *Rhodobacter sphaeroides*, the equivalent experimental substitution also caused a subtle kinetics defect[24]. We propose that, in the *Bcs1l*[p.S78G] mice with insufficient RISP assembly, likely leading to heterodimers with only one active $Q_o$ site, the subtle RISP kinetics defect due to D254N becomes limiting, synergistically decreasing CIII activity. Subsequently, CIII activity drops below the survival threshold, causing lethal metabolic crisis by 5 weeks of age, while *Bcs1l*[p.S78G] mice with WT mtDNA never become as hypoglycemic, escape the early metabolic crisis, and survive to up to 6 months. Intriguingly, we found that *mt-Cyb*[p.D254N] seemed to subtly modify respiratory chain function and metabolism of the mice also in the absence of the *Bcs1l* mutation. For example, *mt-Cyb*[p.D254N] alone was sufficient to increase fatty acid oxidation (low RER) and induce hepatic *Fgf21* expression. The mtDNA sequence data in GenBank show the presence of *mt-Cyb*[p.D254N] in three-toed sloths (*Bradypus*), which feed on an extremely energy-poor diet[16]. It is tempting to speculate that the variant might even be beneficial under some circumstances. Such circumstances might be linked to a need to maximize energy preservation, as alluded by the fact that *mt-Cyb*[p.D254N]-carrying mice had increased electron transfer coupling efficiency, decreased leak respiration, and decreasead nocturnal energy expenditure. Interestingly, in humans the D254N variant (*m.G15506A*) appears in mtDNA sequence database as an extremely rare (3 out of 42,616 full-length mtDNA sequences) polymorphism and is predicted to associate with the African L2b and ancient Central Asian U4b haplogroups (www.mitomap.org).

Lack of methodology to introduce specific mutations into the multicopy mtDNA is still a major limitation in the study of mitochondrial disorders and physiology[32,33]. Other naturally occurring mtDNA variants, carried mostly by outbred mouse strains and of unknown effect on their own, have been bred into various nuclear genetic backgrounds to generate so called conplastic mice. Such strains can be utilized in studies of crosstalk between the mitochondrial and nuclear genomes in the context of metabolism and aging[34,35]. Breeding the *mt-Cyb*[p.D254N] variant into other nDNA backgrounds could offer a novel tool for these studies. Studies in the invertebrate model organisms *Caenorhabditis elegans* and *Drosophila melanogaster* have shown that subtle suppression of mitochondrial respiration can extend life span[36,37]. These findings have remained largely uncorroborated in mammalian models because few viable genetic mouse models with subtle respiratory chain impairment exist[38,39]. Thus the *mt-Cyb*[p.D254N] mice could open up a possibility to test some of the previous seminal findings made in the invertebrate models. In sum, our findings urge further investigations of the metabolic effects of this unique spontaneous, homoplasmic mtDNA variant. Finally, our serendipitous discovery underscores the effect of the genetic background in mouse disease models and encourages replication of phenotyping in a different background to strengthen the biological significance of the findings.

## Methods

**WGS and Sanger sequencing.** Genomic DNA was extracted from frozen liver tissue using standard sodium dodecyl sulfate (SDS)–proteinase K digestion followed by phenol/chloroform extraction and ethanol precipitation. Whole genomes of mice from a C57BL/6JBomTac-derived colony (*n* = 3) and of the C57BL/

6JCrl background (*n* = 2) were sequenced on Illumina HiSeqX platform (National Genomics Infrastructure, SciLifeLab, Sweden) with 30× coverage. Analysis of the sequence data was performed using FastQC, TrimGalore, BWA[40], GATK[41], Bcftools[42], and Annovar ensGene[43] software at Bioinformatics Infrastructure for Life Sciences, Sweden. Mice were genotyped for the *mt-Cyb* variant by Sanger sequencing of a PCR fragment (primers: forward 5′-ACCTCCTCTTCCTCCAC-GAA-3′ and reverse 5′-AGCGTAGAATGGCGTATG CA-3′). Heteroplasmy was evaluated by cloning the *mt-Cyb* amplicon from different tissues into pBluescript plasmid and sequencing bacterial clones.

**Animal experiments and ethics.** *Bcs1l*[c.A232G] knock-in mice in congenic C57BL/6JCrl (Harlan stock 000664) or C57BL/6JBomTac-derived background were maintained in the animal facilities of University of Helsinki, Finland with 12-h light/dark cycle at 22–23 °C. Manual behavioral scoring was used to evaluate the health of homozygous mice as described previously[19] and when the score reached 7/12 the animals were considered to be at end stage of the disease and were euthanized to minimize spontaneous deaths. Genomic DNA was isolated from ear clippings and the mice were genotyped for the *Bcs1l*[c.A232G] mutation as described[8]. The animal experiments were authorized by the national Animal Experiment Board, Finland (ELLA) under the ethical permits ESAVI/6142/04.10.07/2014 and ESAVI/6365/04.10.07/2017 and carried out in accordance with the Federation of Laboratory Animal Science Associations (FELASA) guidelines.

**Sample collection.** The experimental F1 cohort and the mice backcrossed at least thrice to C57BL/6JCrl were fasted for 2 h during the light cycle of the animals before sample collection. The mice were euthanized by carbon dioxide inhalation followed by cervical dislocation. Blood glucose was measured using Freestyle Lite (Abbott, UK) meter. Tissues were immediately collected for histology, mitochondrial isolation, and/or snap-frozen in liquid nitrogen for storage at −80 °C.

**Tissue histology and immunohistochemistry.** Formalin-fixed paraffin-embedded tissues were processed and 5-μm sections stained according to routine procedures for general histology (hematoxylin–eosin (H&E)), glycogen (periodic acid-Schiff (PAS)), and collagen (Sirius Red). For cleaved caspase 3 staining (rabbit monoclonal antibody (mAb) 5A1E, cat. no. #9664, Cell Signaling Technology Inc.), antigen retrieval was performed by boiling the deparaffinized sections for 20 min in 10 mM sodium citrate buffer, pH 6.0. Antibody was diluted 1:1000 and the staining was performed using Vectastain ABC alkaline phosphatase reagents (Vector Laboratories Inc.) and nitroblue teterazolium substrate as per the manufacturer's protocols. Nuclei were counterstained with Nuclear Fast Red (Sigma).

**Quantitative PCR (qPCR).** Total RNA was extracted from snap-frozen liver samples using RNAzol RT reagent (Sigma-Aldrich). RNA purity and concentration were assessed by reading absorbances at 230, 260, and 280 nm and quality by inspecting ribosomal bands after agarose-gel electrophoresis. cDNA was synthesized with RevertAid H− reverse transcriptase (Thermo Scientific, #EP0451) and random hexamers. qPCR reactions comprised Phire Hot-Start II DNA polymerase (Thermo Scientific); EvaGreen DNA-binding dye (Biotum); and buffer, dNTPs, and primers as instructed by the manufacturer of the polymerase. The following primer pairs were used: *Fgf21* (forward, 5′-AGATGGAGCTCTCTATGGATCG-3′ and reverse 5′-GGGCTTCAGACTGGTACACAT-3′), *Gdf15* (forward, 5′-GTCTC CCCGAAGCCTACC-3′ and reverse 5′-TTCAGGGGCCTAGTGATGTC-3′), *Gak* (forward, 5′-CTGCCCACCAGGCATTTG-3′ and reverse 5′-CCATGTCACATAC ATATTCAATGTACCT-3′), and *Rab11a* (forward, 5′-AAGGCACAGATATGG GACACA-3′ and reverse 5′-CCTACTGCTCCACGATAGTATGC-3′). *Gak* and *Rab11a* served as reference genes. CFX96 instrument and CFX Manager software (Bio-Rad) were utilized to perform the qPCR and data analysis.

**Isolation of mitochondria.** Liver and kidney samples were rapidly excised, rinsed of excess blood, minced with scissors, and homogenized in cold isolation buffer (225 mM mannitol, 75 mM sucrose, 10 mM Tris-HCl, 1 mM EGTA, 0.1% bovine serum albumin, pH 7.4 at +4 °C) with glass-teflon Potter-Elvehjem homogenizers. Two-step differential centrifugation (800 × *g* supernatant and 7800 × *g* pellet, 5 min each) was used to obtain a crude mitochondrial preparation for respirometry and Amplex Red-peroxidase assay from the F1 cohort. For other analyses, the isolation was continued by a purification on 19% Percoll (11,300 × *g*, 10 min) followed by washing of the pellet (7800 × *g*, 5 min) with isolation buffer without albumin. Percoll-purified fractions were also used for respirometry from the panel of mice backcrossed to C57BL/6JCrl.

**Homogenization of tissues for enzyme activity measurements.** Snap-frozen heart and skeletal muscle (quadriceps) biopsies were prehomogenized for 15 s with T8 Ultra-Turrax disperser (IKA®-Werke GmbH & Co. KG, Germany) and thereafter homogenized using PBI-Shredder (Pressure Biosciences Inc., MA, USA). Mitochondria isolation buffer served as a homogenization media for the skeletal muscle and 25 mM potassium phosphate buffer (with 40 mM KCl, 1 mM EGTA, pH 7.5) for the heart. We repeated the measurements from the heart and skeletal muscle using samples homogenized with roughened glass-to-glass tissue grinders

(Fig. 2h and Supplementary Fig. 5). Supernatants after low-speed centrifugation (600 × g 10 min at +4 °C) were used for the enzyme activity measurements.

**Assessment of respiratory chain enzymatic activities**. CIII activity was assessed spectrophotometrically by monitoring antimycin A-sensitive cytochrome *c* reduction with decylubiquinol as electron donor in the following reaction mixture: 50 mM potassium phosphate pH 7.5, 60 μM cytochrome *c*, 100 μM decylubiquinol, 2 mM sodium azide, 100 μM EDTA, 0.05% Tween-20 and 0.1 mg/ml bovine serum albumin (BSA). Decylubiquinone was reduced with NaBH₄ and extracted with 2:1 solution of diethylether and isooctane, vacuum dried under nitrogen, and dissolved in acidified EtOH (1 mM HCl). CIII activity data from isolated mitochondria was normalized against mitochondrial protein amount and the data from homogenates to tissue protein or relative mitochondrial mass as assessed by measuring CIV activity. Rotenone-sensitive CI activity was measured with NADH as substrate and decylubiquinone as intermediate electron acceptor coupled to reduction of a colorimetric dye 2,6-dichloroindophenol (DCIP)[44]. Succinate oxidation by CII was coupled to reduction of decylubiquinone and finally DCIP, the absorbance of which was read at 600 nm[45]. CIV activity was measured polarographically with TMPD (*N,N,N',N'*-tetramethyl-p-phenylenediamine dihydrochloride) and ascorbate as electron donors (liver and kidney) or by monitoring cyanide-sensitive cytochrome c oxidation spectrophotometrically (quadriceps and heart). The polarographic assay was corrected for oxygen and sample-dependent TMPD auto-oxidation by measuring residual oxygen consumption at different oxygen levels after addition of sodium azide. All assays were performed at 37 °C.

**Respirometry**. Mitochondrial oxygen consumption was measured in Oxygraph-2k (OROBOROS Instruments, Innsbruck, Austria) chamber at 37 °C in Mir05 buffer (110 mM sucrose, 60 mM lactobionic acid, 20 mM taurine, 20 mM HEPES, 10 mM KH₂PO₄, 3 mM MgCl₂, 0.5 mM EGTA, and 1 g/l fatty acid-free BSA, pH 7.1). Sample, substrates, inhibitors, and uncoupler were injected in the following order: (1) 1 mM malate, 5 mM pyruvate, and 5 mM glutamate; (2) sample; (3) 1.25 mM ADP; (4) 10 μM cytochrome c; (5) 10 mM succinate; (6) 0.5 μg/ml oligomycin A; (7) FCCP titration to maximum respiration; (8) 0.25 μM rotenone; (9) 1 μg/ml antimycin A. Different oligomycin preparations and concentrations (0.5 μg/ml, sc-201551, Santa Cruz vs 0.4 μg/ml, Sigma, O4876) were used for the F1 and later respirometric panels, respectively. The change in oligomycin preparation, a problem discussed by Ruas et al.[46], may explain why we did not observe decreased leak respiration in the liver of *mt-Cyb*^P.D242N^ mice in the F1 data. To normalize any difference in mitochondria mass, we set the maximal capacity of the terminal oxidase, CIV, as the reference state (polarographic CIV activity assay). This in-assay normalization has proven to provide a robust normalization as the measurement is performed immediately after the actual respirometry experiment using same equipment and exactly identical sample amount[47]. Supplementary Fig. 9 shows CI&CII-linked phosphorylating respiration relative to mitochondrial protein.

**Measurement of mitochondrial H₂O₂ production**. Amplex Ultra Red peroxidase assay was used to measure mitochondrial H₂O₂ production simultaneously with respirometry. Oxygraph-2k was equipped with a fluorometer and appropriate filters for resorufin fluorescence (OROBOROS Instruments). The assay reagent concentrations were: 15 μM Amplex Ultra Red (Invitrogen), 1 U/ml horseradish peroxidase, 5 U/ml superoxide dismutase, 10 mM succinate, 1.25 mM ADP, and 0.25 μM rotenone. Horseradish peroxidase-independent Amplex Red oxidation by mitochondrial carboxylesterases[48] was inhibited by 50 μM phenylmethylsulfonyl fluoride. Grossly similar O₂ concentrations were maintained for all samples. The assay was calibrated with a 0.2-nmol bolus of H₂O₂. Stock concentration of H₂O₂ was determined by reading 240 nm absorbance and by employing extinction coefficient of 43.6 M/cm. The background was obtained after addition of catalase (0.5 kU/ml).

**Blue native gel electrophoresis**. Frozen pelleted mitochondria were solubilized using 6 mg digitonin per mg protein in cold solubilization buffer (750 mM aminocaproic acid, 50 mM Bis-Tris, 14 mg/ml digitonin, 1 mM EDTA, pH 7.0 at +4 °C). The solubilized fraction was cleared by centrifugation (18,000 × g, +4 °C, 15 min) and glycerol (5%) and Coomassie Blue G-250 (1.5 mg/ml) were added to the supernatant. Fifteen μg (liver) or 8 μg (kidney) of solubilized mitochondrial protein were resolved in 3–12% NativePAGE^TM^ Bis-Tris gradient gels (Invitrogen) according to the manufacturer's protocol. Following the electrophoresis, the proteins were denatured in 1% SDS for 10 min followed by gel equilibration for 15 min in transfer buffer. Semi-dry electroblotting (25 V, 45 min with ~7 mm interelectrode distance) onto 0.2 μm polyvinylidene difluoride (PVDF) filter was performed using modified Bjerrum Schafer-Nielsen Buffer (48 mM Tris, 39 mM glycine) with cathode buffer supplemented to contain 10% MeOH and 0.05% SDS and anode buffer containing 20% MeOH. After the transfer, background staining was removed using 10% AcOH, 50% MeOH and the transferred proteins were imaged. Complete destaining was accomplished with 100% MeOH before proceeding to immunostaining using the following mAbs from MitoSciences/Abcam: RISP (clone, 5A5), CORE2 (13G12AF12BB11), NDUFA9 (20C11B11B11), ATP5A (15H4C4), and SDHB (21A11AE7). These antibodies were used at concentrations

ranging from 0.2 to 0.3 μg/ml with incubation times ranging from 16 to 20 h at ~8 °C. The primary antibodies were detected using peroxidase-conjugated antibody against mouse IgG (Dako, cat. no. P0047), chemiluminescence, and digital imager (ChemiDoc MP, Bio-Rad). Samples subjected for quantification were loaded onto two 15-well gels in a randomized order. These two gels were simultaneously blotted onto single PVDF membrane, which was thereafter processed as a single unit. The dynamic range of quantification was assessed with a dilution series.

**Indirect calorimetry**. In vivo oxygen consumption and carbon dioxide production were measured using Comprehensive Lab Animal Monitoring System (CLAMS) (Columbus Instruments). The mice were acclimatized in the cages for a minimum of 6 h before the collection of experimental data. This system also recoded movement of the mice.

**Computational methods**. For the crystal structure analyses, structures 1BE3[47], 1L0L[20], 1L0N[27], 1SQV[49], and 2FYU[50] were used. For the MD simulations, a model system of WT enzyme was constructed using the high-resolution (1.9 Å) crystal structure of cytochrome *bc₁* from *Saccharomyces cerevisiae* (PDB: 3CX5)[51]. Detergent molecules, cyt *c*, Q_o site inhibitor stigmatellin, and other crystallographic lipid molecules were removed prior to the modeling set-up. Since cytochrome *bc₁* functions as a homodimer, we incorporated the entire protein structure into a lipid bilayer using CHARMM-GUI[52,53]. The lipid bilayer consisted of three different types of lipids representing the composition of inner mitochondrial membrane, that is, 50.3% POPC (1-palmitoyl-2-oleoyl-sn-glycero-3-phosphocholine) 35.0% POPE (1-palmitoyl-2-oleoyl-sn-glycero-3-phosphoethanolamine), and 14.7% cardiolipin. The membrane–protein set-up was solvated with water molecules and K⁺ and Cl⁻ ions to neutralize the system charge as well as to mimic the 150 mM salt concentration. A similar modeling set-up was applied to construct the mutant (D254N) system (D255 in *S. cerevisiae*). The atomic partial charges of metal cofactors as well as bonded and non-bonded parameters were taken from an earlier study[54]. All redox centers were modeled in their oxidized state. We also performed atomistic MD simulations in the presence of ubiquinone (Q) and ubiquinol (QH₂), for which a QH₂ molecule at the Q_o site and an oxidized Q molecule at the Q_i site, were modeled by aligning to the positions of stigmatellin (PDB: 3CX5) and quinone (Q6 in PDB: 1EZV[55]), respectively. The force field parameters of Q molecules were taken from an earlier study[56]. For all components of the system, CHARMM force field was used[57,58], and simulations were performed with the GROMACS program[59]. The entire system was energy minimized (ca. 6000 steps) using Steepest Descent method available in GROMACS, followed by an equilibration (60 ns). The long production runs (1 μs each for WT and mutant systems, totaling 4 μs) were performed at constant temperature (310 K) and pressure (1 atm), which were established with the use of Nose–Hoover thermostat[60,61] and Parrinello–Rahman barostat[62], respectively. Long-ranged electrostatics was dealt with PME[63] implemented in GROMACS, and LINCS algorithm[64] was used to achieve 2 fs time step. Alignment of trajectories was performed using the protein backbone atoms on to the zeroth frame. The data were plotted as an average from both monomers and also separately (Supplementary Figs. 15 and 16).

**Studies in bacteria**. A mutant strain of *R. capsulatus* containing the substitution of aspartic acid to asparagine on position 278 in cytochrome *b* and a Strep tag at its C-terminus was generated with QuikChange site-directed mutagenesis method (Stratagene) using the pPET1 plasmid (containing WT *petABC* operon) as a template and following PCR primers for mutagenesis (the changed nucleotide in the underlined triplet is in bold):
D278N_F: 5′-CTCGGCCACCCG**A**ACAACTACGTCCAGG-3′ and
D278N-R: 5′-CTGGACGTAGTT<u>G**TT**</u>CGGGTGGCCGAGG-3′.
The *Bst*XI/*Asu*II restriction fragment bearing the introduced mutation was exchanged with its counterpart on a PMTS1-WT plasmid vector and introduced into the MT-RBC1 *R. capsulatus* strain devoid of *petABC* operon using triparental crossing method[65]. The presence of the introduced mutation was confirmed by sequencing the inserted fragment on a plasmid isolated from *R. capsulatus* D278N strain.

*R. capsulatus* strains expressing WT or D278N cytochrome *bc₁* complex were cultured semi-aerobically in MPYE medium. Chromatophores were isolated from pellets of 5-liter bacterial cultures in 50 mM MOPS buffer pH 7.0, 100 mM KCl, and 1 mM EDTA using French press. For cytochrome *bc₁* complex purification, chromatophores were solubilized with n-dodecyl-β-D-maltopyranoside detergent (Anatrace) in 50 mM Tris buffer pH 8.0, 100 mM NaCl, and 1 mM EDTA, ultracentrifuged, and the supernatant was applied on Strep-tag affinity chromatography columns (IBA)[66]. X-band CW EPR spectra of the [2Fe-2S] cluster were measured on Bruker Elexsys E580 spectrometer using Bruker SHQ4122 resonator equipped with an ESR900 cryostat (Oxford Instruments). Measurement parameters were as follows: microwave frequency 9.38 GHz, microwave power 2 mW, modulation amplitude 1.5 mT, modulation frequency 100 kHz, and temperature 20 K.

**Pulse EPR measurements of [2Fe-2S] cluster relaxation**. The temperature dependence of phase relaxation rates of the reduced [2Fe–2S] cluster was measured by pulse EPR spectroscopy in a way similar to that established by Sarewicz

et al.[27,67]. Measurements were carried out on Bruker Elexsys E580 spectrometer using Q-band ER5107D2 resonator inserted into CF935 cryostat. Samples of purified cytochrome $bc_1$ complexes were prepared in bicine buffer pH 8.0 containing 100 mM NaCl, 20% glycerol, and 1 mM sodium ascorbate. Spin echo decay traces were recorded with the use of two pulse sequence ($\pi/2-\tau-\pi$) in which the pulse separation time ($\tau$) was swept. Starting from shortest available $\tau = 160$ ns, the separation time was sequentially incremented by 4 ns interval in 1024 steps. Each time point of the trace was obtained by integration of the spin echo signal appearing after $2\tau$ time. The length of the $\pi/2$ and $\pi$ pulses was set to 16 and 32 ns, respectively, and the microwave power was adjusted to give maximum spin echo signal. Each spin echo decay trace was registered at $g = 1.9$ transition of the [2Fe-2S] cluster spectrum, which for Q-band frequency of 33.6 GHz corresponds to 1264 mT magnetic field induction. Relaxation rates of the reduced [2Fe–2S] cluster were obtained by fitting exponential decay to the electron spin echo traces measured in 12–23 K temperature range.

**Statistics.** Survival data were analyzed with log-rank (Mantel–Cox) test. One-way analysis of variance (ANOVA) and four planned comparisons ($t$ statistics with Welch's correction when appropriate) were performed for normally distributed data: (1) WT vs $Bcs1l^{p.S78G}$, (2) WT vs $mt$-$Cyb^{p.D254N}$, (3) $Bcs1l^{p.S78G}$ vs $Bcs1l^{p.S78G}$; $mt$-$Cyb^{p.D254N}$, and (4) WT and $mt$-$Cyb^{p.D254N}$ vs $Bcs1l^{p.S78G}$ and $Bcs1l^{p.S78G}$;$mt$-$Cyb^{p.D254N}$. The normality assumption of ANOVA was assessed with Shapiro–Wilk method and by manually inspecting the distribution of ANOVA residuals. When required, Kruskal–Wallis and Mann–Whitney $U$ tests were used instead. Analysis of covariance with genotype as a fixed factor and body weight as a covariate was used to calculate weight-normalized energy expenditure. All tests were performed as two-sided tests. The experimental unit ($n$) in the analyses was one individual animal. Throughout the figures, the data are presented as mean and 95% confidence interval of the mean if not otherwise stated. The data points in scatter plots are biological independent replicates.

**Reporting summary.** Further information on research design is available in the Nature Research Reporting Summary linked to this article.

## Data availability

The data are available as Source Data file. Starting structures and input parameters for molecular dynamics simulations are available at https://doi.org/10.5281/zenodo.3383766. We have deposited the $mt$-$Cyb$ variant nucleotide sequence to GenBank (accession MN627229) and WGS data to NCBI Sequence Read Archive (https://www.ncbi.nlm.nih.gov/bioproject/596083), BioSample accessions SAMN13612044, SAMN13612045, SAMN13612046, SAMN13612047, SAMN13612048). Detailed protocols are available from the authors upon request.

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

## Acknowledgements

We thank Dr. Saara Tegelberg, Eva Hansson, Elisa Altay, Rishi Banerjee, and Meharji Arumilli for technical assistance. We thank Dr. Arkadiusz Borek and Dr. Patryk Kuleta for assistance in characterizing the enzymatic activities of the bacterial complexes and Outi Haapanen for help with ubiquinone topology and parameters. The Center for Scientific Computing, Finland is acknowledged for computing time. This study was supported by grants from Academy of Finland (grant 259296 to V.F., grant 294652 to V.S., grant 311031 to N.A.), Swedish Research Council (grant 521-2011-3877 to V.F.), Finnish Physicians' Society (to V.F.), Foundation for Pediatric Research in Finland (to V.F.), Folkhälsan Research Center (to V.F. and J.K.), Magnus Ehrnrooth Foundation (to V.F. and V.S.), University of Helsinki (to V.S.), Sigrid Jusélius Foundation (to V.S.), and National Science Centre, Poland (Maestro grant 2015/18/A/NZ1/00046 to A.O.).

## Author contributions

V.F. and J.K. conceived the project; K.T. performed whole-genome sequence data analysis; J.K. and V.S. MT-CYB protein sequence analyses; V.G. and J.K. *mt-Cyb* genotyping; J.P., V.G. and J.R. mouse behavioral scoring and autopsy; J.P. and V.G. CIII activity assay and respirometry; J.P. Amplex Red-peroxidase assay, qPCR, CI, CII and CIV activity assays, BNGE, and statistics; J.P., V.G. and J.R. tissue histology; R.E., R.P. and A.O. *Rhodobacter* work; and N.A., V.S. and M.W. in silico modeling and simulations. J.P., V.G. and J.K. wrote the manuscript draft and J.P., R.E., N.A., V.S., M.W., A.O., V.F. and J.K. revised it.

## Competing interests

The authors declare no competing interests.
