## [Peer Review File · Nature Communications]

Reviewers' comments:

Reviewer #1 (Remarks to the Author):

The manuscript by Purhonen et al. describes interesting genetic interaction of nuclear and mitochondrial DNA mutations in mitochondrial complex III coding genes (Bcs1l p.S78G and m.14904G>A/Cytb p.D254N, respectively) with the phenotypic consequences being more evident when the two mutations are combined. Given that Cytb p.D254N is rather phenotypically mild (see further), this serendipitous finding highlights the possibility that some OXPHOS defects can arise from converging of two variants that individually are benign (or close to benign). The latter is important for some aspects of diagnosis of respiratory chain defects in humans, therefore, the findings are of interest to others in the community and the wider field.

The paper is well-written in a brief format and the data clearly presented.

However, there are several issues concerning methodology and conclusions the authors should address before publication.

The authors claim that the Mt-Cyb p.D254N mutation in the Bcs1l WT background has phenotypic consequences, but they only show a slight reduction in the weight at P33. Similarly, they also state that the Mt-Cyb p.D254N mutation already has consequences in the Bcs1l WT background in the activity of complex III in liver, but it is very mild (with a lot of inter-individual variability) and it is not the case in any other of the tested tissues. This aspect could be investigated and discussed a bit more thoroughly.

Presentation of activities and bioenergetics data in Figure 2: The way the authors normalize the data could be potentially artefactual and it would be helpful to see the data in a more straightforward way. In Figure 2 (a), CIII activity is normalized by CIV, why? There is no rationale for this. In addition, CIV is measured in two different ways (spectrophotometry and polarography) depending on the tissues, so the data are not really comparable. It would be more correct and consistent if the authors normalized CIII activity by protein (like they do for CI and CII in Supplemental Figure 2) or by citrate synthase (CS) activity, which is the standard method.

It is unclear how the authors normalized the respiration in figures 2b and c. "Relative O₂ flux" seems unconventional? Could the values be expressed as pmol O₂/s/mg of protein. This would also make it easier to compare the values of the old and new backgrounds with the already published data by the same group e.g. in: Leveen et al. 2011. The GRACILE mutation introduced into Bcs1l causes postnatal complex III deficiency: A viable mouse model for mitochondrial hepatopathy. *Hepatology* 53, 437-447 or Davoudi et al. 2014. Complex I Function and Supercomplex Formation Are Preserved in Liver Mitochondria Despite Progressive Complex III Deficiency. *PLoS One* 9, e86767. Here again, there does not seem to be striking differences in the respiratory parameters between the Mt-Cyb genotypes when comparing them within the same Bcs1l genotype.

The functional assays in *Rhodobacter* also indicate that the Mt-Cyb mutation does not have striking consequences in CIII activity. I am sure about the usefulness of the +2Ala mutant shown in figure 4 (h) and (i), this is a drastic mutation that is not found in the mice.

Therefore, the biochemical and functional data shown in the paper indicate a mild additional effect of the Mt-Cyb mutation on top of the Bcs1l mutation. It is therefore a bit unexpected that Mt-Cyb p.D254N accounts for the striking survival phenotype. The manuscript could do much better fairly dealing with this issue.

Lastly, in the previous publications (cited above) the authors demonstrate that there is a CIII activity decrease with age in the Bcs1l mutant mouse in the old background (Mt-Cyb p.D254N). I

wonder if they have checked for the progression of the CIII activity decline with age in the new background (Mt-Cyb WT) to see if it is slower in the critical points and could ameliorate the mouse development and explain the differences in the survival.

Minor point: Nomenclature is not correct Mt-Cyb and MT-CYB (instead of Cytb and CYTB), CYC1 (not CYTC1), RISP is acceptable, but UQCRFS1 is more correct.

Reviewer #2 (Remarks to the Author):

In this work Purhonen et al. have performed in-depth genetic and phenotypic analyses of two different in-house bred mouse strains, both of which harbour the disease-related homozygous p.S78G mutation in the mitochondrial complex III (CIII) assembly factor Bcs1l (involved in the right insertion of the catalytic Rieske FeS protein subunit in CIII), which display either short or long (4-fold) survival rates depending on the C57Bl/6J nuclear genetic (nDNA) background surrounding the p.S78G mutation. WGS analyses identified only eight genetic variants within coding regions to differ between the strains, one of which corresponded to the novel homoplasmic variant m.14904G>A in the mitochondrial DNA (mtDNA) Cytochrome b (Cytb) gene that is exclusively present in the colony with short survival and it affects a negatively charged amino acid (p.D254N) conserved across eukaryotes, making it a likely genetic modifier of the survival of Bcs1l p.S78G mice. To assess the effect of maternally inherited mtDNA to the survival and mitochondrial function of Bcs1l mutant mice, the authors crossbred Bcs1l p.S78G heterozygotes from the two colonies, and compared F1 progenies carrying either wild-type or variant mitochondria with equivalent nDNA background. Using spectrophotometric and polarographic assays they show that the presence of the Cytb variant by itself seems sufficient to induce mitochondrial dysfunction and a metabolic disease phenotype, and that this mutation further exacerbates the CIII activity and mitochondrial respiration defects present in Bcs1l p.S78G mice. In silico simulations predicted that the Cytb variant would compromise the conformational flexibility of the Rieske FeS protein (RISP) head domain necessary for full CIII activity, and the authors show that the equivalent substitution in Rhodobacter Cytb may induce a mild shift in the occupancy of the Rieske head domain towards the quinol oxidation site that, however, neither affects the integrity and activity of Rhodobacter CIII nor bacterial growth.

This is an interesting manuscript where the authors clearly show the epistatic contribution of the Cytb variant m.14904G>A to the mitochondrial CIII defects promoted by the p.S78G mutation in Bcs1l. However, the potential pathogenic role of the Cytb variant alone is unconvincing under the experimental conditions used in this work. First, Bcs1l wild-type mice carrying the Cytb p.D254N variant show similar weight and life span to the wild type mice and histologically, they do not show a clear phenotypic defect. Second, the main phenotypic effect observed in these mice is a significant decrease in CIII activity only in liver tissue (but not in kidney, heart or muscle) which, however, is not reflected in a significant decrease in mitochondrial oxygen consumption in liver (neither C1- nor C1+CII-linked respiration differ significantly, so the use of ratios to demonstrate functional differences here is misleading). The decrease in CIII activity neither correlates with the normal levels/assembly of hepatic complex III observed in these mice. Therefore, how do the authors explain these apparent contradictions? Moreover, as stated by the authors in the Discussion, the Cytb p.D254N variant appears to be present in the human population as a rare polymorphism, suggesting that this situation could occur in mice. To make a clear statement about the action of the maternally-inherited Cytb variant it is advisable to use alternative experimental models easily subject to stress conditions, for instance, cellular cybrids.

The proposed epistatic mechanistic model that converges in the action of the catalytic RISP subunit is very interesting, but it is solely based in computational simulations that lack a convincing experimental demonstration that needs to be provided. As mentioned before, the equivalent Cytb p.D254N variant in Rhodobacter seems to induce a mild defect in the

conformational dynamics of the Rieske head domain that does not alter the integrity and activity of Rhodobacter cIII and therefore, this model is not valid to demonstrate the authors' hypothesis and an alternative experimental model should be presented.

Reviewer #3 (Remarks to the Author):

This paper provides evidence of a double-trouble effect on the biochemical performance of complex III which worsens dramatically the outcome of the Bcs1I mutation associated with the murine equivalent of the GRACILE syndrome, due to the concomitant presence of a missense homoplasmic mutation in the *cytB* gene. The genetic results are convincingly demonstrating that in the presence of the *cytB* mutation the lifespan of the Bcs1I mutant animals falls from about 200 days to 4-5 weeks. This cumulative effect is not caused by obvious structural or assembly defect of complex III in addition to the already severe reduction of complex III amount caused by the Bcs1I mutation. The biochemical characterization in different tissues, mainly liver, kidney and skeletal muscle, demonstrates a cumulative effect in cIII activity in the presence of both mutations (Figure 2a) in liver, kidney and skeletal muscle but not in the heart. The activity of cIII was also slightly but significantly decreased in the liver of Bcs1I-WT, *cytB*-mutant animals, compared to the Bcs1I-WT, *cytB*-WT samples, whereas in other tissues no significant differences were detected. These results suggest a very modest, if any, biochemical effect of the *cytB* mutation when Bcs1I is WT. Other biochemical results require a more detailed explanation, since for instance the normalization procedures are not very clear and justified. For instance, it is unclear what the authors mean by cI-linked and cI-cII linked in figure 2b,c, and what is the meaning of the uncoupling measurement (perhaps to show the maximal respiration rate associated with the different genotypes? This should be explained and the results commented). The same applies to figure 2d (cI&cII/cI), and ETS coupling efficiency (figure 2e). Which is the biological significance of these results in association with the different genotypes considered in the paper. The results shown in figure 2f demonstrate that no increase in H₂O₂ production is associated with the presence of the mutations, alone or in combination with each other. However, the authors mention in the text that the value of H₂O₂ normalized to oxygen consumption is increased in the Bcs1I mutant liver and kidney: does this mean that oxygen is used to produce ROS instead of H₂O because of the block in the ETC? In any case this effect is restricted to Bcs1I mutants whereas no effect is mentioned attributable to the mutation in *cytB*. To test (and confirm) that the two mutations are epistatically linked, the authors made *in silico* prediction and experiments based on spectroscopic measurements on the phase relaxation of RISP Fe-S cluster by oxidized heme bL. I am not familiar with this technology and I must rely on the conclusions of the authors who conclude that the *cytB* mutation stiffens *cytB* and interferes with the interaction between *cytB* and the Rieske protein, worsening the electron flux in a cIII mutant characterized by severe reduction of RISP incorporation, due to the Bcs1I mutation. The conclusions of the authors are that this is an example of synergistic effects of two mutations, one in the nuclear genome, the other in mtDNA, in determining a substantial worsening of the phenotype linked to cIII deficiency. The existence of causative correlations between mtDNA mutations/polymorphisms with each other has been demonstrated in humans (e.g. for LHON mutations). The possibility of a cumulative effect between a mtDNA mutation and the impairment of a nuclear gene has been also hypothesized, for instance in the ND6 mouse mutant associated with a deletion in the NNT gene. However, the present paper indicates an etiological link of these effects based on structural interactions. The explanation in the present case is based mainly on the experiments on *Rhodobacter capsulatus* (It is unclear what the 2Ala mutant is...), whereas the mutation in *cytB* has otherwise no biochemical or clinical phenotype. I am not sure that this experimental setup provides a persuasive explanation about the worsening of the phenotype. The *in silico* predictions suggest an increased "stiffness" of *cytB*, which can slightly alter the motion of *cytB* and, possibly, of the RISP. This effect has no consequences in the Bcs1I-WT, whereas it causes a very dramatic worsening of the clinical phenotype, which is disproportionately severe compared to the differences in the cIII activities, O₂ consumption, and other biochemical measurements displayed in Fig. 2.

Reviewer #4 (Remarks to the Author):

This paper reports a Cytochrome b variant (Cytb^{p.D254N}) that further decreases CIII activity and capacity of mitochondrial respiration in Bcs1l^{p.S78G} mice. Molecular dynamics simulations of *S. cerevisiae* cyt bc1 and variant Cytb^{p.D254N} indicate that the observed *in vivo* differences can be attributed to the compromised mobility of the EF loop and to the Iron-Sulfur Protein Subunit Extrinsic Domain -ISP-ED (or RISP head domain). The mutation was also experimentally tested in *R. capsulatus* (E278N) where no significant differences to the enzyme activity were observed in relation to the WT. A subtle effect was detected on the motion of ISP-ED towards the Qo site.

Rajagukguk, S. *et al.* previously described this mutation in *Rhodobacter sphaeroides* (D278N) (Biochemistry. 2007; 46(7): 1791–1798). The novelty of the present work thus resides in the combination of Cytb^{p.D254N} with Bcs1l^{p.S78G}. The effect of D254N alone is very subtle, as showed in the present work and previously by Rajagukguk, S. *et al.*, where only a small change in the rate constant for electron transfer from the iron-sulfur center [2Fe2S] to cyt c1 was found (35,000 s⁻¹ for D278N and 60,000 s⁻¹ for the WT enzyme). The MD simulations are state of the art and extensive enough (1μs) to provide adequate sampling. However, the authors should describe in more detail the modelled complex. The model was built from the crystal structure 3CX5, which also includes cytochrome c and the inhibitor stigmatellin, which arrests ISP-ED movement. The inhibitor was obviously deleted, but it seems that the Qo- and Qi -sites were left empty since no information is given about the substrates quinol (QH₂) or quinone (Q). If that is the case the simulations do not describe any intermediate of the catalytic cycle (Q-cycle). This represents a missing opportunity to study the effect of mutations in the actual catalytic cycle. Due to the proximity of the Qo-site to the EF loop it would be interesting to analyse the effect of the substrate on the EF loop dynamics. The authors should clarify this point.

Finally, taking in account that the effect of the mutation is subtle and since the enzyme is a homodimer, the authors should compare the RMSFs of each monomer.

Minor points:

It is stated that :

“The atomic partial charges of metal cofactors were taken from an earlier study 49”

I suppose the authors wanted to mention that the atomic partial charges and remaining parameters were taken from ref. 49. If the equilibrium bonds, angle and van der Waals parameters were taken from somewhere else that should be stated in the manuscript

Below, please find our detailed response (in purple font) and a list of references supporting the response. In the response, the main changes and additions to the manuscript text and figure panels are referred to by figure and line numbers, respectively, and highlighted in red font.

Reviewer #1 (Remarks to the Author):

The manuscript by Purhonen et al. describes interesting genetic interaction of nuclear and mitochondrial DNA mutations in mitochondrial complex III coding genes (*Bcs1l*p.S78G and m.14904G>A/*Cytb*.D254N, respectively) with the phenotypic consequences being more evident when the two mutations are combined. Given that *Cytb* p.D254N is rather phenotypically mild (see further), this serendipitous finding highlights the possibility that some OXPHOS defects can arise from converging of two variants that individually are benign (or close to benign). The latter is important for some aspects of diagnosis of respiratory chain defects in humans, therefore, the findings are of interest to others in the community and the wider field. The paper is well-written in a brief format and the data clearly presented. However, there are several issues concerning methodology and conclusions the authors should address before publication.

The authors claim that the Mt-*Cyb* p.D254N mutation in the *Bcs1l* WT background has phenotypic consequences, but they only show a slight reduction in the weight at P33. Similarly, they also state that the Mt-*Cyb* p.D254N mutation already has consequences in the *Bcs1l* WT background in the activity of complex III in liver, but it is very mild (with a lot of inter-individual variability) and it is not the case in any other of the tested tissues. This aspect could be investigated and discussed a bit more thoroughly.

The major revelation of our study is the serendipitous and infinitesimally unlikely identification of a novel spontaneous mtDNA variant that has a drastic effect when combined with a known disease-causing *Bcs1l* mutation. As for the effect of the D254N alone, we have to stress the fact that the mice carrying this spontaneous variant are healthy “wild-type” mice that have been used by us (see references in manuscript) and several other research groups for about a decade without any suspicion. Therefore, we did not expect D254N to have any measurable effect on its own, and we consider the significantly lower weight and decreased liver CIII activity in the juvenile F1 mice a striking finding. For this revision, we have measured the hepatic gene expression of *Fgf21* and *Gdf15*, two important metabolic regulators and secreted markers of mitochondrial dysfunction. Interestingly, *mt-Cyb*^{p.D254N} alone induced the expression of *Fgf21* to a similar level as the pathogenic *Bcs1l* mutation, indicating that it is sufficient to cause a clear metabolic stress signal. Importantly, the *Fgf21* upregulation was maintained also in mice backcrossed several times to C57BL/6JCrI (Fig. 1f and Results, lines 83-88).

The fact that D254N alone does not seem to decrease CIII activity in all tissues is not unexpected and is in line with the extensive literature showing tissue specific effects (pleiotropy) of both mtDNA mutations and nuclear mutations affecting mitochondria (Frazier et al., 2017). This is, in fact, the case for human *BCS1L* and *MT-CYB* mutations as well. For example, different *BCS1L* mutations cause very different clinical pictures, sometimes with mainly visceral manifestations (as in the GRACILE syndrome caused by the S78G mutation, which primarily affect the liver and the kidney) and sometimes no visceral manifestations at all (Fellman and Kotarsky, 2011; Fernández-Vizarra and Zeviani, 2015) (Discussion, lines 260-265).

Concerning the inter-individual variability in liver CIII activity data (Fig. 2a, n=5/genotype), we already replicated the liver CIII activity measurement from the F1 *Bcs1l* WT mice with and without *mt-Cyb^{p.D254N}* (n=8/genotype) and these data was shown in the Supplementary Fig 2d. We have now moved this replication data to main Fig. 2b. The heart CIII activity data did indeed show high inter-individual variability that likely originated from the unrepresentatively small (5 mg) frozen tissue samples we originally used. We have now optimized the sample preparation (sample amount, buffer and homogenization; **Materials and Methods, lines 352-358**) and repeated the CIII activity measurement for the heart. This new data clearly shows the synergistic effect of *Bcs1l^{p.S78G}* and *mt-Cyb^{p.D254N}* on CIII activity also in this tissue (Fig. 2a). Interestingly, the new data with less variation also suggests that *mt-Cyb^{p.D254N}* has an independent effect on CIII activity in the heart.

To rule out type 1 error regarding the independent effect of *mt-Cyb^{p.D254N}* on liver and heart CIII activity, we have now bred an additional mouse panel that is genetically somewhat different from the original C57BL/6JBomTac:C57BL/6JCrI F1 hybrid mice, as these mice have been backcrossed to C57BL/6JCrI males several times as part of the normal maintenance of the colony. We collected a sample panel from these mice at approximately P30, which is a few days earlier than in the original F1 panel and repeated the liver and heart CIII activity measurements. Importantly, the measurements confirmed the synergistic decrease in CIII activity in the *Bcs1l^{p.S78G};mt-Cyb^{p.D254N}* mice after the further backcrossing to C57BL/6JCrI (**Supplementary Fig. 2**). However, the nuclear background does seem to have a slight effect as the hepatic CIII activity was not significantly decreased by *mt-Cyb^{p.D254N}* alone in this panel. In the heart, however, *mt-Cyb^{p.D254N}* decreased CIII activity also in the *Bcs1l* WT mice. We have added these data in the supplement (**Supplementary Fig. 2**).

We have also revised the text throughout, better pointing out the matters presented above.

Presentation of activities and bioenergetics data in Figure 2: The way the authors normalize the data could be potentially artefactual and it would be helpful to see the data in a more straightforward way. In Figure 2 (a), CIII activity is normalized by CIV, why? There is no rationale for this. In addition, CIV is measured in two different ways (spectrophotometry and polargography) depending on the tissues, so the data are not really comparable. It would be more correct and consistent if the authors normalized CIII activity by protein (like they do for CI and CII in Supplemental Figure 2) or by citrate synthase (CS) activity, which is the standard method.

We measured CIII activities for liver and kidney from isolated mitochondria and normalized the activities to protein concentration as stated in the text, and now also more clearly in the figure legend. We only collected frozen heart and skeletal muscle tissue because it is technically possible to measure CIII activities from homogenates of these tissues (unlike e.g. liver) and because we did not want to isolate mitochondria from these tissues due to poor yield from small amounts of tissues (small P30-35 mice). With the homogenates (in contrast to isolated mitochondria), however, normalization to protein concentration gave unacceptable within-group variation. Therefore, we normalized the CIII activity data to the unaffected mitochondrial inner-membrane enzyme, CIV, for these two tissues. Our validation procedure for any normalization method is to first confirm that the normalization is not biased by the experimental set up (e.g. by genotype). Second, we check that the normalization decreases the within-group variation and correlates (within-group) with the target to be normalized. We observed similar CIV activity in all genotypes, and normalization to CIV activity did not alter relative mean values between the groups. It, however, efficiently decreased the within-group variation and correlated with CIII activity (average within-group $R^2 = 0.87$ for CIII vs CIV correlation). We agree that the CS activity is a widely used marker and it can be an excellent marker for mitochondrial mass (Larsen et al. J Physiol 2012). We, however, found that CS activity

poorly correlated with CIV ($R^2=0.24$) or CIII activity ($R^2=0.23$) in our heart samples. Normalization of heart CIII activity to CS activity actually increased the within-group coefficient of variation from 31% to 42% whereas normalization to CIV activity decreased the coefficient of variation to 13.6%. Moreover, we have previously shown that in *Bcs1l* mutant mice skeletal muscle CS activity can be affected by a sole dietary modification without obvious increase in mitochondrial mass (Purhonen et al., 2017). Larsen et al. (J Physiol 2012) found that in human muscle biopsies, among 13 markers of mitochondrial mass, CIV activity correlated third best, after cardiolipin content and CS activity, with mitochondrial mass. In sum, we think that normalization to CIV activity for these two tissues is a valid method. We have now clarified our normalization method (**Materials and Methods, lines 365-369**).

It is unclear how the authors normalized the respiration in figures 2b and c. “Relative O₂ flux” seems unconventional? Could the values be expressed as pmol O₂/s/mg of protein. This would also make it easier to compare the values of the old and new backgrounds with the already published data by the same group e.g. in: Leveen et al. 2011. The GRACILE mutation introduced into *Bcs1l* causes postnatal complex III deficiency: A viable mouse model for mitochondrial hepatopathy. *Hepatology* 53, 437-447 or Davoudi et al. 2014. Complex I Function and Supercomplex Formation Are Preserved in Liver Mitochondria Despite Progressive Complex III Deficiency. *PLoS One* 9, e86767. Here again, there does not seem to be striking differences in the respiratory parameters between the Mt-Cyb genotypes when comparing them within the same *Bcs1l* genotype.

Analysis of mitochondrial function by respirometry requires fresh samples and due to the low-throughput nature of respirometry we had to spread the experiment over a few weeks. To minimize interday variability, we chose to normalize the data to CIV activity that can be measured inside the respirometry chamber using TMPD and ascorbate as substrates immediately after all other steps. In other words, the maximal oxygen consumption capacity, by the terminal oxidase CIV, was set as a reference state. This strategy provides a robust in-assay normalization as the measurement is performed immediately after the actual respirometry experiment using the same equipment and exactly identical sample amount (Larsen et al., 2012). For the initial respirometry experiments, we used a simple two-step centrifugation that provides a crude but minimally processed, intact mitochondrial fraction. As the respirometry data is central in this study, we have now repeated the whole respirometry experiment with a new independent mouse panel. We also used more purified mitochondrial fraction that allowed us to measure the mitochondrial protein amount more accurately. We have now presented this data both as percentage of maximal CIV capacity (**Supplementary Fig. 4**) and relative to mitochondrial protein (**Supplementary Fig. 5**). These new data faithfully replicate our previous measurements. We have now clarified this normalization method to the reader (**Materials and Methods, lines 386-395**).

Standardization of respirometry data is an important issue in the field. Unfortunately, currently, comparisons of absolute respiration values between different laboratories is difficult. Respirometry for the Leveen et al. 2011 article was performed a long time ago in a different laboratory, by different personnel, a slightly different protocol and using mice of 129Sv:C57BL/6J mixed background. Thus, we do not find comparing absolute respiration values between these two studies meaningful.

The functional assays in *Rhodobacter* also indicate that the Mt-Cyb mutation does not have striking consequences in CIII activity. I am sure about the usefulness of the +2Ala mutant shown in figure 4 (h) and (i), this is a drastic mutation that is not found in the mice.

As the reviewer correctly points out, D278N indeed does not have an effect on CIII activity in the purified *Rhodobacter bc₁* complex. This is in line with the subtle effect it has alone in mice (it is not a disease-causing mutation). As we mention in the Discussion, only a slight kinetics defect was previously shown in another bacterial model, *Rhodobacter sphaeroides*, when this mutation was introduced as an artificial experimental mutation (Rajagukguk et al., 2007). We utilized the +2Ala mutant as a positive reference and to illustrate a more prominent effect on the RISP mobility. We find that showing the +2Ala mutant data in Fig. 4 for comparison is useful to estimate the magnitude of the effect of the D254N mutation. +2Ala is an artificial mutation and the rationale of its use is now explained better in the text (Results, lines 192-194).

Therefore, the biochemical and functional data shown in the paper indicate a mild additional effect of the Mt-Cyb mutation on top of the Bcs1l mutation. It is therefore a bit unexpected that Mt-Cyb p.D254N accounts for the striking survival phenotype. The manuscript could do much better fairly dealing with this issue.

Like explained above, the effect of D254N is necessarily subtle because the mice carrying it are basically healthy wild-type mice. In the *Bcs1l* mutant mice, however, even a slight further decrease in CIII activity can be detrimental. Of the 20 or so known BCS1L mutations, the S78G mutation causes the most severe phenotype in human patients, likely leading to the severest CIII deficiency possible without embryonic lethality (knock-out alleles of respiratory chain subunits, including RISP, are almost always embryonic lethal). The mice carrying the analogous mutation are not quite as sick, surviving to P200, probably because they avoid an early lethal metabolic crisis, but they are still very sick and growth-retarded from weaning on. We have now elaborated the discussion, better bringing forward this point.

Lastly, in the previous publications (cited above) the authors demonstrate that there is a CIII activity decrease with age in the *Bcs1l* mutant mouse in the old background (Mt-Cyb p.D254N). I wonder if they have checked for the progression of the CIII activity decline with age in the new background (Mt-Cyb WT) to see if it is slower in the critical points and could ameliorate the mouse development and explain the differences in the survival.

Yes, we think the reviewer's point is highly relevant. The *Bcs1l* mutant phenotype first manifests soon after weaning (approximately P24). In the *mt-cyb^{p.D254N}* background this coincides with 50% loss of CIII activity, and the activity in liver drops linearly to as low as 10% just prior to death (Levéen et al., 2011 and the current manuscript). Although we lack a similar time series from this age window in the *mt-Cyb^{wt}* background, our current data and previously measured CIII activities at P95 (Purhonen et al. 2017) and at end-stage P200 (Rajendran et al. under review) are approximately as follows: P30: 40%, P33: 25%, P95: 30% and P200: 35% of wild-type littermate values. In sum, these data show that in the *mt-cyb^{p.D254N}* background CIII activity rapidly declines to 10% by P35, resulting in lethal metabolic crisis (blood glucose <2), whereas in the *mt-Cyb^{wt}* background the activity has a nadir (~25%) at approximately the same age but stabilizes or even slightly recovers thereafter. The plain *Bcs1l^{p.S78G}* homozygotes never develop lethal metabolic crisis but die of late-onset (>P150) dilating cardiomyopathy (Rajendran et al. under review). Again, our crossbreeding experiment rules out the effect of nuclear background on the difference in the course of CIII activity decline, indicating that it is determined by the *mt-cyb^{p.D254N}* variant alone.

Minor point: Nomenclature is not correct Mt-Cyb and MT-CYB (instead of Cytb and CYTB), CYC1 (not CYTC1), RISP is acceptable, but UQCRFS1 is more correct.

Corrected.

Reviewer #2 (Remarks to the Author):

In this work Purhonen et al. have performed in-depth genetic and phenotypic analyses of two different in-house bred mouse strains, both of which harbour the disease-related homozygous p.S78G mutation in the mitochondrial complex III (CIII) assembly factor Bcs1l (involved in the right insertion of the catalytic Rieske FeS protein subunit in CIII), which display either short or long (4-fold) survival rates depending on the C57Bl/6J nuclear genetic (nDNA) background surrounding the p.S78G mutation. WGS analyses identified only eight genetic variants within coding regions to differ between the strains, one of which corresponded to the novel homoplasmic variant m.14904G>A in the mitochondrial DNA (mtDNA) Cytochrome b (Cytb) gene that is exclusively present in the colony with short survival and it affects a negatively charged amino acid (p.D254N) conserved across eukaryotes, making it a likely genetic modifier of the survival of Bcs1l.p.S78G mice. To assess the effect of maternally inherited mtDNA to the survival and mitochondrial function of Bcs1l mutant mice, the authors crossbred Bcs1l.p.S78G heterozygotes from the two colonies, and compared F1 progenies carrying either wild-type or variant mitochondria with equivalent nDNA background. Using spectrophotometric and polarographic assays they show that the presence of the Cytb variant by itself seems sufficient to induce mitochondrial dysfunction and a metabolic disease phenotype, and that this mutation further exacerbates the CIII activity and mitochondrial respiration defects present in Bcs1l p.S78G mice. In silico simulations predicted that the Cytb variant would compromise the conformational flexibility of the Rieske FeS protein (RISP) head domain necessary for full CIII activity, and the authors show that the equivalent substitution in Rhodobacter Cytb may induce a mild shift in the occupancy of the Rieske head domain towards the quinol oxidation site that, however, neither affects the integrity and activity of Rhodobacter CIII nor bacterial growth.

“This is an interesting manuscript where the authors clearly show the epistatic contribution of the Cytb variant m.14904G>A to the mitochondrial CIII defects promoted by the p.S78G mutation in Bcs1l. However, the potential pathogenic role of the Cytb variant alone is unconvincing under the experimental conditions used in this work. First, Bcs1l wild-type mice carrying the Cytb p.D254N variant show similar weight and life span to the wild type mice and histologically, they do not show a clear phenotypic defect. Second, the main phenotypic effect observed in these mice is a significant decrease in CIII activity only in liver tissue (but not in kidney, heart or muscle) which, however, is not reflected in a significant decrease in mitochondrial oxygen consumption in liver (neither CI- nor CI+CII-linked respiration differ significantly, so the use of ratios to demonstrate functional differences here is misleading).”

We actually do not claim that *mt-Cyb^{p.D254N}* is pathogenic, and therefore we refer to it as a variant, not as a mutation. Indeed, the mice carrying it have been used as healthy wild-types by us and several other research groups for years. Therefore, we were very surprised that our results showed subtle changes in growth and respiratory chain function in these mice as compared to mice carrying wild-type C57BL mitochondria. Mitochondrial defects commonly produce highly tissue-specific phenotypes and even many severely pathogenic mutations do not usually manifest in every tissue.

In many tissues CIII activity has to decline at least by 50% for a pathological phenotype to arise. A subtle CIII defect may not limit CI&CII-linked respiration in all tissues and cell types, depending on their metabolic status and rate. Ratios of respiratory states have been used since the very first respirometry experiments (Chance and Williams, 1955). They provide an internal normalization and, thus are more sensitive than absolute respiration values to pick subtle differences (Pesta and

Gnaiger, 2012). We disagree that the use of ratios is misleading. They provide a functional fingerprint of the electron transport system and, in our case, show that the *mt-Cyb*^{p.D254N} variant does modify the mitochondrial function. We have now elaborated the text and explained the parameters more thoroughly. We have also added data on oligomycin-induced leak respiration which was significantly decreased by the D254N variant (Fig. 3a, b and Supplementary Fig. 5 and 6).

“The decrease in CIII activity neither correlates with the normal levels/assembly of hepatic complex III observed in these mice. Therefore, how do the authors explain these apparent contradictions?”

Indeed, our conclusion from BNGE analysis of CIII composition/assembly was that D254N did not affect CIII assembly, i.e. the amount of RISP assembled into CIII, so as to explain the decrease in CIII activity. We reasoned that its effect has to be more subtle, outside the sensitivity of any measurements we could do using isolated mitochondria. Therefore, we decided to perform molecular dynamics simulations and functional studies in an organism (*Rhodobacter capsulatus*), in which the cytochrome bc₁ complex can be mutagenized and purified. These two additional approaches both showed that the defect lies in RISP dynamics. Therefore, we do not see any contradiction.

“Moreover, as stated by the authors in the Discussion, the *Cytb* p.D254N variant appears to be present in the human population as a rare polymorphism, suggesting that this situation could occur in mice. To make a clear statement about the action of the maternally-inherited *Cytb* variant it is advisable to use alternative experimental models easily subject to stress conditions, for instance, cellular cybrids.”

The *mt-Cyb*^{p.D254N} variant may theoretically exist in mice as a rare polymorphisms that was amplified by random drift in the colony we used, but in any case, as mentioned in the manuscript, it is not present in any *Mus musculus* mtDNA sequence available in GenBank.

The robust *in vivo* stress condition that revealed the presence of this variant in the first place was the *Bcs1l* mutation compromising CIII function. In other words, the *mt-Cyb*^{p.D254N} carriers were much more sensitive to the effect of the *Bcs1l*^{p.S78G} mutation than mice with wild-type mitochondria. It is fully possible that the mice are more sensitive to other genetic, environmental or chemical stress factors affecting or dependent on mitochondrial function, e.g. other mutations or chemicals inhibiting the respiratory complexes, but we feel that such further *in vivo* studies are outside the scope of this manuscript.

The reviewer is right in that cybrid cell lines are an established model to assess the effect of mtDNA mutations. However, in our case, a long history of studies of patient fibroblasts shows that CIII deficiency tends to manifest very mildly if at all in cell lines, and this is the case for the *BCS1L*^{p.S78G} mutation as well, even though the patients are otherwise extremely sick already at birth (Fellman and Kotarsky, 2011; Fellman et al., 1998; Kotarsky et al., 2010; Visapää et al., 2002). Recently, a *MT-CYB* missense mutation (*m.15557G>A*, E271K), causing a drastic change of a negatively charged amino acid into a positive one in an extremely conserved sequence motif was modelled in cybrid cells. This mutation, predicted to dramatically compromise CIII function, induced only a mild mitochondrial dysfunction in human transmitochondrial cybrids (Iommarini et al., 2018). In sum, we think that the probability of *mt-Cyb*^{p.D254N} cybrid cells producing meaningful additional data is too small to justify their time-consuming generation for this study. We agree that cybrids may be of interest in further studies if it is feasible to generate such from cell types relevant for the disease pathology, such as a hepatocyte cell line.

The proposed epistatic mechanistic model that converges in the action of the catalytic RISP subunit

is very interesting, but it is solely based in computational simulations that lack a convincing experimental demonstration that needs to be provided. As mentioned before, the equivalent *Cytb* p.D254N variant in *Rhodobacter* seems to induce a mild defect in the conformational dynamics of the Rieske head domain that does not alter the integrity and activity of *Rhodobacter* CIII and therefore, this model is not valid to demonstrate the authors' hypothesis and an alternative experimental model should be presented.

In fact, the epistatic mechanism we propose is not based solely on computational simulations, but on them combined with assessment of CIII function in intact isolated mitochondria (respirometry), mitochondrial membranes (CIII enzymatic activity) and extensive analysis of purified *Rhodobacter bc₁* complex. The mouse crossbreeding experiment showed unequivocally that the maternally inherited *Cytb*^{p.D254N} variant dictated the survival difference between the two colonies. Our further unpublished work on these mice has shown that the survival difference is stable and remains the same after further backcrossing of the *Cytb*^{p.D254N} carriers to C57BL/6JCrI males. We show that the variant decreased CIII activity in key affected tissues (liver and kidney) and in skeletal muscle and heart of *Bcs1l* mutant mice. The results from structural modelling, molecular dynamics simulations and analyses of the purified *Rhodobacter bc₁* complex, showing a subtle effect on *ef* loop and RISP motility, are of the magnitude we expected because D254N is not a disease-causing mutation (it is present in a "wild-type" mouse colony). Compared with the various disease-causing patient and artificial mutations previously simulated and assayed in the *Rhodobacter* model (Borek et al., 2015; Ekiert et al., 2016; Lee et al., 2011), we did not expect the D254N variant to cause a severe defect. Nevertheless, to corroborate the data from the F1 mice, we have now bred another experimental mouse panel using parental mice that have been backcrossed to C57BL/6JCrI for several times (as opposed to the original C57BL/6JBomTac:C57BL/6JCrI F1 data presented in the manuscript) and repeated the respirometry and CIII activity assays from this independent panel, with almost identical results (Supplementary Figs. 2, 4 and 5).

The *Rhodobacter* model has been widely used and accepted as a model system to study effects of mitochondrial mutations in complex III due to high homology of the core subunits between bacterial and eukaryotic cytochromes *bc₁*. See examples in (Borek et al., 2015; Lanciano et al., 2013). The reviewer does not elaborate which alternative model he/she would advise us to use, but the only eukaryotic model organism into which mtDNA mutations can be introduced artificially is yeast. While we do acknowledge the general importance of corroborating results in several experimental models, we find it highly unlikely that studying the D254N variant in isolated yeast mitochondria would provide further insight as compared to mouse mitochondria and the purified *Rhodobacter* enzyme that we have used in this study. It is theoretically implausible that any other mechanism than the novel mtDNA variant could account for the maternally inherited phenotypic difference between the two *Bcs1l* mutant strains. The only X chromosomal variant (in *Rhox6* gene, see Supplementary Table 1) we identified by WGS is synonymous (Leu51Leu) and the crossbreeding experiment rules out also a sex-specific effect. In sum, we feel that the data we present from three different experimental models is coherent and sound, and that an additional experimental model is not required.

Reviewer #3 (Remarks to the Author):

"This paper provides evidence of a double-trouble effect on the biochemical performance of complex III which worsens dramatically the outcome of the *Bcs1l* mutation associated with the murine equivalent of the GRACILE syndrome, due to the concomitant presence of a missense homoplasmic mutation in the *cytB* gene. The genetic results are convincingly demonstrating that in the presence of the *cytB* mutation the lifespan of the *Bcs1l* mutant animals falls from about 200

days to 4-5 weeks. This cumulative effect is not caused by obvious structural or assembly defect of complex III in addition to the already severe reduction of complex III amount caused by the Bcs11 mutation. The biochemical characterization in different tissues, mainly liver, kidney and skeletal muscle, demonstrates a cumulative effect in CIII activity in the presence of both mutations (Figure 2a) in liver, kidney and skeletal muscle but not in the heart. The activity of CIII was also slightly but significantly decreased in the liver of Bcs11-WT, cytB-mutant animals, compared to the Bcs11-WT, cytB-WT samples, whereas in other tissues no significant differences were detected. These results suggest a very modest, if any, biochemical effect of the cytB mutation when Bcs11 is WT. Other biochemical results require a more detailed explanation, since for instance the normalization procedures are not very clear and justified. For instance, it is unclear what the authors mean by cI-linked and cI-cII linked in figure 2b,c, and what is the meaning of the uncoupling measurement (perhaps to show the maximal respiration rate associated with the different genotypes? This should be explained and the results commented). The same applies to figure 2d (CI&CII/cI), and ETS coupling efficiency (figure 2e). Which is the biological significance of these results in association with the different genotypes considered in the paper.”

We have now thoroughly revised the Results text concerning respirometry to be more reader-friendly (Results, lines 110-145). By CI-linked respiration we mean respiration driven by substrate combination providing NADH for CI. Similarly, CI&CII-linked respiration is respiration driven by CI-linked substrates and CII substrate, succinate. In some tissues (e.g. mouse liver) the phosphorylation system (adenine nucleotide translocase, phosphate transporters and the ATP synthase) limits respiration, thus we also measured uncoupled respiration. As mentioned in the response to Reviewer #1, technical replication and the new data obtained with a new mice cohort shows that *mt-Cyb^{p.D254N}* has an independent effect on CIII activity also in the heart.

“The results shown in figure 2f demonstrate that no increase in H₂O₂ production is associated with the presence of the mutations, alone or in combination with each other. However, the authors mention in the text that the value of H₂O₂ normalized to oxygen consumption is increased in the Bcs11 mutant liver and kidney: does this mean that oxygen is used to produce ROS instead of H₂O because of the block in the ETC? In any case this effect is restricted to Bcs11 mutants whereas no effect is mentioned attributable to the mutation in cytB.”

We interpret that higher H₂O₂ production per O₂ consumption means higher relative leakage of electrons to oxygen before the terminal oxidase, CIV that reduces O₂ to produce H₂O.

“To test (and confirm) that the two mutations are epistatically linked, the authors made in silico prediction and experiments based on spectroscopic measurements on the phase relaxation of RISP Fe-S cluster by oxidized heme bL. I am not familiar with this technology and I must rely on the conclusions of the authors who conclude that the cytB mutation stiffens cytB and interferes with the interaction between cytB and the Rieske protein, worsening the electron flux in a CIII mutant characterized by severe reduction of RISP incorporation, due to the Bcs11 mutation. The conclusions of the authors are that this is an example of synergistic effects of two mutations, one in the nuclear genome, the other in mtDNA, in determining a substantial worsening of the phenotype linked to CIII deficiency. The existence of causative correlations between mtDNA mutations/polymorphisms with each other has been demonstrated in humans (e.g. for LHON mutations). The possibility of a cumulative effect between a mtDNA mutation and the impairment of a nuclear gene has been also hypothesized, for instance in the ND6 mouse mutant associated with a deletion in the NNT gene. However, the present paper indicates an etiological link of these effect based on structural interactions. The explanation in the present case is based mainly on the experiments on *Rhodobacter capsulatus* (It is unclear what the 2Ala mutant is...), whereas the

mutation in *cytB* has otherwise no biochemical or clinical phenotype. I am not sure that this experimental setup provides a persuasive explanation about the worsening of the phenotype. The *in silico* predictions suggest an increased "stiffness" of *cytB*, which can slightly alter the motion of *cytB* and, possibly, of the RISP. This effect has no consequences in the *Bcs11*-WT, whereas it causes a very dramatic worsening of the clinical phenotype, which is disproportionately severe compared to the differences in the CIII activities, O₂ consumption, and other biochemical measurements displayed in Fig. 2."

The *Bcs11* mutant mice have severe CIII deficiency and are sick from early life, so it is not at all unexpected that even a subtle further decrease in CIII function may dramatically worsen the disease progression via a threshold effect. In humans, different *BCS1L* mutations are extremely pleiotropic, i.e. cause an unexpectedly wide spectrum of phenotypes even with apparently similar CIII deficiency. Thus, in the light of the literature on mitochondrial disorders and our own work, we do not find the biochemical measurements disproportional in relation to the phenotype observed in the mice.

The +2Ala mutant in *Rhodobacter* work was used as a reference to illustrate the more prominent effect of arresting RISP movement on the temperature-dependent phase relaxation to compare its magnitude with the effect of D278N mutation. +2Ala is an artificial mutation and the rationale of its use is now explained better in the text.

We do, in fact, clearly show in the current work that the *mt-Cyb* variant alone has biochemical and metabolic consequences: significantly lower weight, decreased liver and heart CIII activity, lower leak respiration and altered ratios of different respiratory states. Additionally, for the revision, we have measured the hepatic gene expression of *Fgf21* and *Gdf15*, two important metabolic regulators and secreted markers of mitochondrial dysfunction. Interestingly, *mt-Cyb*^{p.D254N} alone induced the expression of *Fgf21* to a similar level as the pathogenic *Bcs11* mutation, indicating that it is sufficient to cause a clear metabolic stress signal (Fig. 1f and Results, lines 83-88).

The *mt-Cyb* variant dramatically affects survival of the *Bcs11* mutant mice. We do not claim that it dramatically worsens the overall clinical phenotype. It is impossible to determine for sure their immediate cause of death, but the most likely cause is lethal metabolic crisis due to extreme hypoglycemia (blood glucose commonly below detection limit, Fig. 1e), which the plain *Bcs11* mutant mice escape. In other words, the worsening is directly related to energy deficiency due to the CIII defect. Otherwise, the organ manifestations (liver and kidney disease), which progress fairly slowly, are similar between the colonies, with only proteinuria clearly more pronounced in the kidneys of the *Bcs11*^{p.S78G};*mt-Cyb*^{p.D254N} mice (Supplementary Fig. 1).

Reviewer #4 (Remarks to the Author):

This paper reports a Cytochrome b variant (*Cytb*^{p.D254N}) that further decreases CIII activity and capacity of mitochondrial respiration in *Bcs11*^{p.S78G} mice. Molecular dynamics simulations of *S. cerevisiae* *cyt bc1* and variant *Cytb*^{p.D254N} indicate that the observed *in vivo* differences can be attributed to the compromised mobility of the EF loop and to the Iron-Sulfur Protein Subunit Extrinsic Domain -ISP-ED (or RISP head domain). The mutation was also experimentally tested in *R. capsulatus* (E278N) where no significant differences to the enzyme activity were observed in relation to the WT. A subtle effect was detected on the motion of ISP-ED towards the Qo site. Rajagukguk, S. *et al.* previously described this mutation in *Rhodobacter sphaeroides* (D278N) (Biochemistry. 2007; 46(7): 1791–1798). The novelty of the present work thus resides in the combination of *Cytb*^{p.D254N} with *Bcs11*^{p.S78G}. The effect of D254N alone is very subtle, as showed in the present work and previously by Rajagukguk, S. *et al.*, where only a small change in the rate

constant for electron transfer from the iron-sulfur center [2Fe2S] to cyt c1 was found (35,000 s⁻¹ for D278N and 60,000 s⁻¹ for the WT enzyme).

We thank the reviewer for pointing out this important detail. Indeed, in agreement with this earlier data as well as the data presented in this work, we find the effect of the D254N mutant to be subtle also in the simulations. We have now performed additional simulations as suggested by the reviewer and amended the manuscript text accordingly (Supplementary Figs. 6, 8-9).

The MD simulations are state of the art and extensive enough (1 μ s) to provide adequate sampling. However, the authors should describe in more detail the modelled complex. The model was built from the crystal structure 3CX5, which also includes cytochrome c and the inhibitor stigmatellin, which arrests ISP-ED movement. The inhibitor was obviously deleted, but it seems that the Q_o- and Q_i -sites were left empty since no information is given about the substrates quinol (QH₂) or quinone (Q). If that is the case the simulations do not describe any intermediate of the catalytic cycle (Q-cycle). This represents a missing opportunity to study the effect of mutations in the actual catalytic cycle. Due to the proximity of the Q_o-site to the EF loop it would be interesting to analyse the effect of the substrate on the EF loop dynamics. The authors should clarify this point.

We have now added a more detailed description of the modeling setup, including the modeling of quinone molecules at the Q_o and Q_i sites. We performed additional ~1 μ s simulations of wild-type and mutant systems each by modeling doubly reduced, doubly protonated QH₂ and an oxidized Q at the Q_o and Q_i sites, respectively (Supplementary Fig. 6). As the reviewer pointed out, we indeed find that the modeling of QH₂ molecule partly arrests the flexibility of the domains around the site D254, which is only 11-13 Å from the Q_o site, and somewhat more in the case of D254N mutant, in agreement with the current and earlier biochemical data.

Finally, taking in account that the effect of the mutation is subtle and since the enzyme is a homodimer, the authors should compare the RMSFs of each monomer.

We have now added new simulations (Supplementary Figs. 8, 9) showing data from the individual monomers and also discussed the differences observed in between the monomers in more details.

Minor points: It is stated that: “The atomic partial charges of metal cofactors were taken from an earlier study 49”. I suppose the authors wanted to mention that the atomic partial charges and remaining parameters were taken from ref. 49. If the equilibrium bonds, angle and van der Waals parameters were taken from somewhere else that should be stated in the manuscript

We have now clarified this in the manuscript.

References

- Borek, A., Kuleta, P., Ekiert, R., Pietras, R., Sarewicz, M., and Osyczka, A. (2015). Mitochondrial disease-related mutation G167P in Cytochrome b of *Rhodobacter capsulatus* Cytochrome bc₁ (S151P in Human) affects the equilibrium distribution of [2Fe-2S] cluster and generation of superoxide. *J. Biol. Chem.* 290, 23781–23792.
- Ekiert, R., Borek, A., Kuleta, P., Czernek, J., and Osyczka, A. (2016). Mitochondrial disease-related mutations at the cytochrome b-iron-sulfur protein (ISP) interface: Molecular effects on the large-

- scale motion of ISP and superoxide generation studied in *Rhodobacter capsulatus* cytochrome bc₁. *Biochim. Biophys. Acta* 1857, 1102–1110.
- Fellman, V., and Kotarsky, H. (2011). Mitochondrial hepatopathies in the newborn period. *Semin. Fetal. Neonatal Med.* 16, 222–228.
- Fellman, V., Rapola, J., Pihko, H., Varilo, T., and Raivio, K.O. (1998). Iron-overload disease in infants involving fetal growth retardation, lactic acidosis, liver haemosiderosis, and aminoaciduria. *Lancet Lond. Engl.* 351, 490–493.
- Fernández-Vizarra, E., and Zeviani, M. (2015). Nuclear gene mutations as the cause of mitochondrial complex III deficiency. *Front. Genet.* 6, 134.
- Frazier, A.E., Thorburn, D.R., and Compton, A.G. (2017). Mitochondrial energy generation disorders: genes, mechanisms and clues to pathology. *J. Biol. Chem.* pii: jbc.R117.809194.
- Iommarini, L., Ghelli, A., Leone, G., Tropeano, C.V., Kurelac, I., Amato, L.B., Gasparre, G., and Porcelli, A.M. (2018). Mild phenotypes and proper supercomplex assembly in human cells carrying the homoplasmic *m.15557G>A* mutation in *CYTOCHROME B* gene. *Hum. Mutat.* 39, 92–102.
- Kotarsky, H., Karikoski, R., Mörgelin, M., Marjavaara, S., Bergman, P., Zhang, D.-L., Smet, J., van Coster, R., and Fellman, V. (2010). Characterization of complex III deficiency and liver dysfunction in GRACILE syndrome caused by a *BCS1L* mutation. *Mitochondrion* 10, 497–509.
- Lanciano, P., Khalfaoui-Hassani, B., Selamoglu, N., Ghelli, A., Rugolo, M., and Daldal, F. (2013). Molecular mechanisms of superoxide production by complex III: A bacterial versus human mitochondrial comparative case Study. *Biochim. Biophys. Acta* 1827, 1332–1339.
- Larsen, S., Nielsen, J., Hansen, C.N., Nielsen, L.B., Wibrand, F., Stride, N., Schroder, H.D., Boushel, R., Helge, J.W., Dela, F., et al. (2012). Biomarkers of mitochondrial content in skeletal muscle of healthy young human subjects. *J. Physiol.* 590, 3349–3360.
- Lee, D.-W., Selamoglu, N., Lanciano, P., Cooley, J.W., Forquer, I., Kramer, D.M., and Daldal, F. (2011). Loss of a conserved tyrosine residue of Cytochrome b induces reactive oxygen species production by Cytochrome bc₁. *J. Biol. Chem.* 286, 18139–18148.
- Levéen, P., Kotarsky, H., Mörgelin, M., Karikoski, R., Elmér, E., and Fellman, V. (2011). The GRACILE mutation introduced into *Bcs1l* causes postnatal complex III deficiency: a viable mouse model for mitochondrial hepatopathy. *Hepatol.* 53, 437–447.
- Pesta, D., and Gnaiger, E. (2012). High-resolution respirometry: OXPHOS protocols for human cells and permeabilized fibers from small biopsies of human muscle. *Methods Mol. Biol.* 810, 25–58.
- Purhonen, J., Rajendran, J., Mörgelin, M., Uusi-Rauva, K., Katayama, S., Krjutskov, K., Einarsdottir, E., Velagapudi, V., Kere, J., Jauhiainen, M., et al. (2017). Ketogenic diet attenuates hepatopathy in mouse model of respiratory chain complex III deficiency caused by a *Bcs1l* mutation. *Sci. Rep.* 7, 957.
- Rajagukguk, S., Yang, S., Yu, C.-A., Yu, L., Durham, B., and Millett, F. (2007). Effect of mutations in the cytochrome b *ef* loop on the electron-transfer reactions of the Rieske iron-sulfur protein in the cytochrome bc₁ complex. *Biochemistry* 46, 1791–1798.

Visapää, I., Fellman, V., Vesa, J., Dasvarma, A., Hutton, J.L., Kumar, V., Payne, G.S., Makarow, M., Van Coster, R., Taylor, R.W., et al. (2002). GRACILE syndrome, a lethal metabolic disorder with iron overload, is caused by a point mutation in *BCS1L*. *Am. J. Hum. Genet.* *71*, 863–876.

Reviewers' comments:

Reviewer #1 (Remarks to the Author):

Thanks again for allowing me having another look into the manuscript by Purhonen et al. In their revision the authors have provided incremental improvements to the previously presented work, however, many aspects have not been addressed satisfactory.

The authors have brought FGF21 into play to support the notion of MT-CYTB p.D254N having a very minor phenotypic consequence in the BCS11 WT background. They do see a slight increase in this marker between BCS11 WT +MT-CYTB p.D254N, but there is no increase in FGF21 when MT-CYTB D254N is present in the BCS11 p.S78G background. For GDF15 the overall pattern is different, yet again MT-CYTB D254N+ BCS11 S78G does not elevate the levels of this marker as compared to having one of these mutations separately. This all leaves the reader wondering why the MT-CYTB D254N+BCS11 S78G mice are so dramatically different as compared to MT-CYTB WT+BCS11 S78G. This new experiment does not clarify this point at all.

The comment about the presentation of RCC activities was virtually ignored. It is still not clear why CIII activity could not be measured in low spin homogenate across the tissues – the method which is commonly used in the field.

Similarly to the FGF21 and GDF15, the respiration data still do not allow explaining why MT-CYTB D254N+BCS11 S78G mice are so dramatically affected as compared to MT-CYTB p.D254N+ BCS11 WT or MT-CYTB WT+BCS11 S78G strains.

Reviewer #2 (Remarks to the Author):

Given the new evidences provided, the conclusion that the subtle Cytb variant m.14904G>A epistatically contributes to the mitochondrial CIII defects promoted by the p.S78G mutation in the assembly factor Bcs1l remains unconvincing, as other different genetic causes (both homozygous and heterozygous, the latter not being contemplated at all by the authors, as well as potential mutations outside coding regions) cannot be excluded as causative of the defect (as also indicated by the authors in their rebuttal letter). Moreover, the bacterial model presented to support the functional influence of such mild mutation is inadequate. The detrimental effect of the cytb mutation by itself needs to be convincingly demonstrated, as the effects shown are very mild (in the best case) or non-existent and the calculations of the statistical significance are based on a very limited number of measurements per experiment. Since the authors have obtained strains that are equalized in nDNA and differ in mtDNA, they could effectively demonstrate the functional influence of the cytb mutation by stressing MEFs or fibroblasts by metabolic switch, among other different experimental approaches. The authors need to convincingly demonstrate the detrimental effect of the mutation in cytb by itself (as it is satisfactorily shown in the paper by Iomarinni et al mentioned by the authors in their rebuttal letter, among others). They could alternatively make cybrids to ensure that the detrimental effect comes exclusively from the mutation detected in the mtDNA since without a clear proof, any other genetic variant exclusively present in the variant-carrying strain can be equally responsible for the phenotypic effects described in this work. In addition, there is no clear correlation between complex III activities, O₂ consumption assays and complex III assembly levels. The respiratory chain activity of complex III cannot be normalized by complex IV activity as this complex does not reflect mitochondrial mass. Even if the authors claim that there are no differences in complex IV activities between strains, complex IV is often affected by defects of complex III and it cannot be used for normalization at the authors' will. If the normalization by standard procedures such as citrate synthase or by milligrams of protein results in increased inter individual variability, this probably reflects the lack of significant differences among sample measurements. The same goes for the use of ratios and the non-

standard procedures for the normalization of O₂ consumption experiments. In this regard, I am particularly worried by the method used to reach such great statistical significance given the small number of measurements provided and the clear overlapping error bars in most of the experiments.

In summary, my major concerns remain unsatisfactorily addressed.

Reviewer #3 (Remarks to the Author):

The revised paper has resolved some of the questions I raised in my comments. The work, especially the genetic results, is convincing in showing a synergistic double-trouble effect of a virtually benign polymorphism in cyt b vs. a severe mutation in the UQCRFS1 assembly factor Bcs1l. As I mentioned in my previous comments, this is an interesting finding although the presence of these effects is not an absolute novelty in the specification of mitochondrial phenotypes. The mechanistic explanation offered by the Authors is interesting and supported by some experimental data obtained in a mtDNA-editable organism such as *R. capsulatus*.

Reviewer #4 (Remarks to the Author):

In my opinion the authors significantly improved the manuscript by adding to the revised version two additional 1 μ s simulations of the protein with the substrates ubiquinone and ubiquinol modeled at the Q_o and Q₁ sites and by analyzing the differences between the monomers. There are still some minor issues with the paper. The labels in the new figures (S8 and S9) are misleading. It seems that the authors are referring to different simulations, when in fact the graphs are for the two monomers. So instead of wild-type 1 and wild-type 2, I would suggest something like WT monomer 1, WT monomer 2.

Furthermore, and since the effects reported in the simulations are subtle, I suggest changing "Accordingly, we observed a simultaneous reduction" to "a simultaneous small reduction" (Page 8, line 170)

Finally, the distance between the dihydroxyquinone in the modeled structures and D254/N254 is too large for any direct interaction. So the sentence "this is in part due to the vicinity of the QH2 molecule at the Q_o site to the D254/N254 residue (average distance from the simulations between the Q head group and the CG atom of D254/N254 is $\approx 13.2/11.6$ Å)" should be deleted and instead the authors should analyze the distance(s) between other of loop residue(s) or nearby residues that are closer to QH2. (Page 8, line 177).

Reviewer #5 (Remarks to the Author):

The ms thoroughly analysis the effect of a mutant of the bc1 complex in mice, and propose it is due to a restricted mention of the Rieske protein, as a result of a D254N mutation.

In spite of the impressive amount of work performed, all data regarding the activity of the bc1 mutant, including the work performed with a similar mutant in the complex from the bacterium *R. capsulatus*, show only, and this are the authors' words, subtle effects, if any at all. Therefore, this raises a key question – is this mutant really important in causing diseases, when the activity of the complex is barely affected? Also, the molecular dynamics simulations reveal only a very minor effect on the dynamics of the Rieske protein, when the calculations are performed upon docking the quinone substrates.

I am not an expert on Pulsed EPR, but at least the EPR spectra of the WT and mutant complexes

are also identical.

Other specific points are:

- Lines 117-118 – If the activity of the bc1 were affected, how the absence of effect on CI-linked respiration increased? It should have happened exactly the opposite, if what the authors claim is really important
 - Lines 121-122 – Again, CI-CII linked phosphorylation is not affected by the mutation, neither maximal electron transfer capacity, which should have been observed if the D254N mutation would be relevant. (lines 141-142).
 - Lines 147-152 – ROS production is also not affected (also in the *R. capsulatus* mutant).
 - In fact, the MD simulations contradict the results – while it is claimed, on its basis, that the mobility of the Rieske protein is affected, this does not translate into an effect on its activity!
- In summary, all biochemical and respirometry data clearly show that the D254N mutation has indeed no effect on the mitochondrial respiration of the mice mutants. Therefore, some other reasons must exist to explain the phenotypes observed (and those, only in some tissues).

Purhonen et al. A spontaneous cytochrome b variant exacerbates complex III
deficiency in mice by restricting Rieske Fe-S protein motion

To all reviewers:

We initiated this multi-approach study after observing a consistent, extraordinary 5-fold difference
in the survival of CIII deficient *Bcs1l* mutant mice between two colonies of different strains (in
Lund, Sweden and in Helsinki, Finland). Therefore, we performed whole genome sequencing of
both strains and were stunned to discover a previously unknown mtDNA variant in the Lund
colony, in which the homozygotes had a short survival. Amazingly, of the astronomically high
number possible nuclear and mitochondrial genetic variants that could theoretically modify
mitochondrial function and energy metabolism (and subsequently the complex multiorgan disease
phenotype), the variant we identified alters a subunit of CIII, the same complex compromised due
to the *Bcs1l* mutation. Even more stunningly, the variant is located in the binding site for the
electron-transferring subunit, RISP, the assembly of which is compromised by the *Bcs1l* mutation.
As opposed to nuclear DNA, mtDNA is inherited only maternally. Therefore, it only took a single
F1 crossbreeding experiment from the two inbred mouse lines (mice homozygous for every allele)
to produce F1 hybrid mice (all nDNA differences in heterozygous state) and prove that the short
survival trait is maternally inherited and, thus, must necessarily be caused by the only difference in
mtDNA, *mt-Cyb*^{p.D254N}. Affecting the mtDNA-encoded CIII subunit MT-CYB, around which the
whole mammalian 11-subunit CIII is assembled, and which functions as the core structural and
catalytic subunit, it was also obvious and theoretically inevitable that the only way this variant
could aggravate the mouse phenotype is by further compromising CIII function.

**Reviewer #1 (Remarks to the Author):**

Thanks again for allowing me having another look into the manuscript by Purhonen et al. In their
revision the authors have provided incremental improvements to the previously presented work,
however, many aspects have not been addressed satisfactory.

The authors have brought FGF21 into play to support the notion of MT-CYTB p.D254N having a
very minor phenotypic consequence in the BCS11 WT background. They do see a slight increase in
this marker between BCS11 WT +MT-CYTB p.D254N, but there is no increase in FGF21 when
MT-CYTB D254N is present in the BCS11 p.S78G background. For GDF15 the overall pattern is
different, yet again MT-CYTB D254N+ BCS11 S78G does not elevate the levels of this marker as
compared to having one of these mutations separately. This all leaves the reader wondering why the
MT-CYTB D254N+BCS11 S78G mice are so dramatically different as compared to MT-CYTB
WT+BCS11 S78G. This new experiment does not clarify this point at all.

The main novelty of this study is that we identify the genetic cause for the drastic survival
difference between the two mouse colonies. We have already characterized the mutant phenotypes
in both mouse colonies separately (see references in the manuscript) during the past 10 years,
including their CIII activity and assembly and energy metabolism. In the Lund colony, the
homozygotes die due to metabolic crisis involving extreme hypoglycemia by P40. The appearance
and progression of their disease correlates with linear post-weaning loss of CIII activity. (Leveen et
al 2011, Kotarsky et al. 2012, Davoudi et al. 2014, Davoudi et al. 2016, Rajendran et al. 2016,
Purhonen et al. 2017). The long-living Helsinki colony, derived by embryo transfer from the Lund
colony and backcrossing to C57BL/6JCr1, has likewise been characterized in several publications
(Purhonen et al. 2017, Tegelberg et al. 2017, Purhonen et al. 2018, Rajendran et al. 2018). These

mice never become as hypoglycemic as the Lund mice and live to P200 because they escape the
early metabolic crisis. We have now clarified these premises of the current study in the manuscript.

We agree with the reviewer that the *Fgf21* and *Gdf15* mRNA expression data does not explain the
phenotypic difference, but that was not our hypothesis either. The purpose of this data was to
strengthen the assessment of the possible metabolic effect of the *mt-Cyb*^{D254N} variant alone, as
suggested by the reviewers previously. There was no particular reason to presume that in our CIII
deficiency model *Fgf21* or *Gdf15* expression should have been further increased in the livers of the
variant-carrying mutant mice or that these factors contribute to their more rapid metabolic
deterioration. However, it is quite remarkable that hepatic *Fgf21* expression was upregulated by the
non-pathogenic *mt-Cyb*^{D254N} variant alone to similar degree as in the sick *Bcs1l*^{p.S78G} mice. These
data are coherent with our other data showing subtle alteration of respiratory chain function, and, in
this second revision, also by our indirect calorimetry data showing that the variant is non-
pathogenic but not functionally silent.

The comment about the presentation of RCC activities was virtually ignored. It is still not clear why
CIII activity could not be measured in low spin homogenate across the tissues – the method which
is commonly used in the field.

We apologize for not answering directly to the reviewer's initial proposal of presenting the data
relative to protein amount. These methods have not been standardized across different laboratories,
especially not among non-clinical laboratories such as ours. Many technical details of the assay
such as choice of protein standard and assay chemistry may already introduce two-fold difference in
"absolute" values. Thus, we found it more informative to present the CIII activity data relative to
healthy control group as this allows inspections of threshold effects (e.g. 50% and 25% of wild-type
values) across different tissues and time points.

We agree that it is possible to measure activity of several mitochondrial enzymes, such as succinate
dehydrogenase and citrate synthase activity, from homogenates across different tissues. However, it
is stated in the literature (Spinazzi et al. 2012 and Medja et al. 2009) that liver homogenates are not
amenable for measurement of specific activity of CIII, and isolated mitochondria are required.

Similarly to the FGF21 and GDF15, the respiration data still do not allow explaining why MT-
CYTB D254N+BCS11 S78G mice are so dramatically affected as compared to MT-CYTB
p.D254N+ BCS11 WT or MT-CYTB WT+BCS11 S78G strains.

We have now clarified throughout the manuscript that the mice are dramatically sicker because their
CIII activity decreases below survival threshold (25%) in liver and below 50% of wild-type values
(threshold for pathology) in other assessed tissues due to *mt-Cyb*^{p.D254N}, as we show in Fig. 2. This
further aggravation of CIII deficiency in multiple tissues could easily collapse the whole-body
metabolism, as indeed our new indirect calorimetry data suggests (Fig. 6).

The purpose of the respirometry analyses was to assess how the *mt-Cyb*^{D254N} might further
exacerbate the well-characterized (see references in manuscript) respiration defect due to mutated
BCS1L. CIII deficiency as such is difficult to measure by respirometry, and may or may not
significantly manifest with different sample types and substrate combinations. We did two
independent replications, and the *mt-Cyb*^{p.D254N} carrying *Bcs1l* mutant mice showed consistent
further decrease in maximal phosphorylating respiration, as assessed by using convergent electron
flow to the coenzyme Q pool via CI and CII (CI&CII-linked OXPHOS). To further strengthen the
data on OXPHOS defect, we have now measured hepatic ATP levels at various time points. These
measurements show a linear age-dependent decrease in hepatic ATP content in *Bcs1l*^{p.S78G}; *mt-*
*Cyb*^{p.D254N} mice (Fig. 1f). A few days before the expected death the hepatic ATP concentration was
approximately one quarter of WT values. This time point coincided with near-exponential increase
in hepatocyte apoptosis (Fig. 1g). In contrast, the *Bcs1l*^{p.S78G} mice with wild-type *mt-Cyb* showed

milder ATP depletion and even some recovery by P35-36, and never developed massive hepatocyte
 apoptosis.

**Reviewer #2 (Remarks to the Author):**

Given the new evidences provided, the conclusion that the subtle *Cytb* variant m.14904G>A
 epistatically contributes to the mitochondrial CIII defects promoted by the p.S78G mutation in the
 assembly factor *Bcs1l* remains unconvincing, as other different genetic causes (both homozygous
 and heterozygous, the latter not being contemplated at all by the authors, as well as potential
 mutations outside coding regions) cannot be excluded as causative of the defect (as also indicated
 by the authors in their rebuttal letter).

It is unfortunate that we failed to present and explain clearly enough in the manuscript and in the
 first response the crucial genetic (crossbreeding) experiment (Fig. 1b and c) that shows
 unequivocally that the short survival was inherited maternally. The only maternally inherited
 genetic material is mtDNA, and the only difference in mtDNA between the two colonies was the
 *mt-Cytb*^{D254N} variant we identified by WGS. This reciprocal crossbreeding experiment excluded the
 effect of homozygous nuclear variants as they all necessarily become heterozygous in the F1
 progeny. Heterozygous variation as the cause is essentially impossible because the C57BL/6 mouse
 strains are inbred (homozygous for nearly all alleles), and the short survival consistently manifests
 in 100% of *Bcs1l*^{p.S78G} homozygotes. Therefore, *mt-Cytb*^{D254N} is explicitly the cause of the survival
 difference. The rest of the data from the mice are observations about how the variant modifies the
 tissue histopathology, CIII function and respiration in *Bcs1l* mutant mice, and have no bearing on
 the conclusion from the genetic experiment. For the first revision, we additionally used mice that
 underwent embryo transfer and backcrossing. In these mice, the short survival remained exclusively
 maternally inherited, further ruling out any nuclear genetic variant as a significant contributing
 factor. Because this fundamental and most important piece of evidence was obviously not presented
 clearly enough by us, we have now revised several parts of the manuscript. See also response to
 Reviewer 1.

Moreover, the bacterial model presented to support the functional influence of such mild mutation
 is inadequate.

We have now revised the manuscript to explain more clearly that *mt-Cytb*^{D254N} is a non-pathogenic
 variant (rather than a mutation), present in healthy “wild-type” mice. We employed the
 *Rhodobacter* model because it allows expression of genetically modified cytochrome b from a
 plasmid and purification of the very simple 3-component bacterial bc₁ complex for investigation of
 the subtle structural changes predicted by the structural modelling and molecular dynamics
 simulations. The spectroscopy analyses confirmed the predicted subtle effect, exactly as we
 expected on the basis of the location and nature of the mutation in the *ef* loop, as well as the
 simulations. Therefore, the bacterial model was the only possible system to test the computational
 predictions, showing the effect of the variant at atomic level.

The detrimental effect of the *cytb* mutation by itself needs to be convincingly demonstrated, as the
 effects shown are very mild (in the best case) or non-existent and the calculations of the statistical
 significance are based on a very limited number of measurements per experiment.

Again, it seems that we have failed to express clearly that *mt-Cytb*^{p.D254N} is a non-pathogenic variant
 present in a wild-type mouse colony and, therefore, it should not have a detrimental effect on its
 own. Yet, this does not imply that it is completely silent. Indeed, we show that the variant had a
 slight but significant metabolic effect (as non-pathogenic mtDNA variants, i.e. haplotypes, have
 been suggested to have in human populations). The group sizes we used are based on 3R principles
 of animal experimentation, our own previous data (Leveen et. al 2011, Davoudi et al. 2014,

Davoudi et al. 2016, Rajendran et al. 2016, Purhonen et al. 2017, Purhonen et al. 2018, Rajendran et
al. 2018), power calculations and literature on mouse studies of respiratory chain deficiency. For
example, a 20% further decrease in CIII activity in CIII deficient mice is physiologically a robust
effect, and can easily account for the appearance of lethal metabolic crisis. For the first revision, we
also reassessed the main parameters from an independent second mouse panel with similar results.

Since the authors have obtained strains that are equalized in nDNA and differ in mtDNA, they
could effectively demonstrate the functional influence of the *cytb* mutation by stressing MEFs or
fibroblasts by metabolic switch, among other different experimental approaches. The authors need
to convincingly demonstrate the detrimental effect of the mutation in *cytb* by itself (as it is
satisfactorily shown in the paper by Iomarinni et al mentioned by the authors in their rebuttal letter,
among others).

We thank the reviewer for bringing up the idea to utilize a cell culture model, which we did in fact
consider during the project. Iomarrini et al. studied a *MT-CYB* mutation that was predicted to
dramatically effect CIII function. Despite this, they found only a mild phenotype in cybrid cells.

CIII deficiency generally does not manifest in fibroblasts, which is the primary cell type easily
available from patients and mice. We have shown this for the human *BCS1L*^{S78G} mutation, causing
the most severe CIII deficiency phenotype in patients (Kotarsky et al. 2010), as well as for another
compound heterozygous mutation (Tegelberg et al. 2017). Therefore, we deemed it very unlikely
that the subtle *mt-Cyb*^{D254N} variant would manifest as altered CIII activity or respiration in a cell
line. In the current manuscript, we already provide substantial *in vivo* evidence that *mt-Cyb*^{D254N}
does have a significant metabolic effect on its own (cardiac CIII activity, leak respiration, energy
expenditure and *Fgf21* expression).

They could alternatively make cybrids to ensure that the detrimental effect comes exclusively from
the mutation detected in the mtDNA since without a clear proof, any other genetic variant
exclusively present in the variant-carrying strain can be equally responsible for the phenotypic
effects described in this work.

As acknowledged by the reviewer in the previous comment, we obtained mice that have equalized
nDNA and differing in mtDNA, and because the *mt-Cyb*^{p.D254N} variant was the only difference in the
mtDNA, other genetic variants have been ruled out.

In addition, there is no clear correlation between complex III activities, O₂ consumption assays and
complex III assembly levels.

We regret that our figure panels did not highlight the most important pieces of the data well enough.
We previously provided additional respirometry data from the backcrossed mice as a supplement,
which we have now moved to a main figure for clarity. We have now also plotted the correlation
between the CIII activity and maximal phosphorylating respiration, and it is actually almost perfect.
See graph below for the liver data, also added as Supplementary Fig. 8.

The fact that the respiration driven by sole NADH-producing substrates (CI-linked respiration) is
 not compromised by the CIII deficiency is in line with our previous published work and with
 studies on threshold effects related to respiratory enzymes. For instance, rat liver CIII has to be
 inhibited by more than 85% before CI-linked respiration is affected (Rossignol et al. 1999).
 Therefore, to reveal CIII deficiency in respirometry one has to use convergent electron flow to the
 coenzyme Q pool via at least two quinol oxidoreductases such as CI and CII (CI&CII-linked
 respiration).

The second fact is that the uncoupled respiration did not separate the genotypes as significantly as
 the phosphorylating respiration. Interpretation of this respiratory state is, however, quite
 complicated as CIII is a major proton translocase in the respiratory electron transfer and the
 uncoupling may shift the rate-limiting steps in the system. It has been shown that the activity of
 uncoupled CIII activity is approximately 3 times higher than when it is working against membrane
 potential (Rich and Clark 1982, Rottenberg et al. 2009). Moreover, the accurate estimation of
 maximal respiration by uncoupling is somewhat technically challenging due to different tolerances
 of healthy and compromised mitochondria to protonophores and off-target effects of oligomycin
 preparations (uncoupling after oligomycin) (Ruas et al. 2016).

In summary, the most important piece of respirometry data is the quasi-maximal phosphorylation
 respiration as estimated by measuring CI&CII-linked phosphorylation respiration. We have now
 made major revisions to the figure panels to clarify this most important piece of data.

References not included in manuscript:

Rich PR, Clarke SD. Reconstitution of cytochrome bc1 complex into lipid vesicles and the restoration of uncoupler
 sensitivity. FEBS Lett 148:54-8 (1982)

Rossignol, R., Malgat, M., Mazat, J. P. & Letellier, T. Threshold effect and tissue specificity. Implication for
 mitochondrial cytopathies. J Biol Chem 274, 33426-33432 (1999)

Rottenberg H, Covian R, Trumpower BL. Membrane Potential Greatly Enhances Superoxide Generation by the
 Cytochrome bc1 Complex Reconstituted into Phospholipid Vesicles. J Biol Chem 2009;284(29):19203-10.

Ruas, J. S. et al. Underestimation of the maximal capacity of the mitochondrial electron transport system in oligomycin-
 treated cells. Plos One 11 (2016).

The respiratory chain activity of complex III cannot be normalized by complex IV activity as this
 complex does not reflect mitochondrial mass.

Larsen et al (2012) showed that the correlation coefficient for CIV activity and mitochondrial mass
 in human biopsy samples is around 0.8. Of course, CIV deficiencies and mtDNA maintenance
 defects are likely to be an exception where CIV activity might not correlate with mitochondrial

mass. This is, however, not the case with our CIII deficient mice. Normalization to CIV activity can
 be interpreted as a normalization to respiratory chain content, which we think is actually more
 relevant in our case than mitochondrial mass.

Even if the authors claim that there are no differences in complex IV activities between strains,
 complex IV is often affected by defects of complex III and it cannot be used for normalization at the
 authors' will. If the normalization by standard procedures such as citrate synthase or by milligrams
 of protein results in increased inter individual variability, this probably reflects the lack of
 significant differences among sample measurements. The same goes for the use of ratios and the
 non-standard procedures for the normalization of O₂ consumption experiments.

We apologize for not highlighting this clearly, but we actually did provide the respirometry data
 relative to mitochondrial proteins as a supplement in the first revision. The conclusions were
 unaffected. Moreover, liver and kidney CIII activity data are normalized to mitochondrial protein.

We find that it is better to normalize the data to the CIII-CIV segment of the respiratory chain by
 measuring CIV activity. More accurately, this strategy provides normalization to the respiratory
 chain content and not just to mitochondrial mass. In respirometry, CIV sets the maximum oxygen
 consumption capacity, which provides a meaningful reference point and highly sensitive in-assay
 normalization. By analogy, if we were studying a TCA cycle enzyme, then it would make more
 sense to normalize to citrate synthase activity that is part of the TCA cycle. In our previous
 response, we explained more thoroughly the rationale and provided supporting literature for the
 normalization. The only plausible mechanism how an amino acid change in a central electron-
 transferring CIII subunit can modify the phenotype of CIII deficient mice is by affecting CIII
 structure and/or function, and the purpose of CIII activity measurements and respirometry is to
 show the magnitude of the effect.

In this regard, I am particularly worried by the method used to reach such great statistical
 significance given the small number of measurements provided and the clear overlapping error bars
 in most of the experiments.”

Please, see also comment (lines 137-143) above. We thank the reviewer for paying attention to the
 statistical methodology, the correctness of which is a very important part of reproducible high-
 quality science. Instead of the standard error of the mean (SEM) combined with a bar graph
 presentation, we follow by default published recommendations for data presentation (for example,
 Krzywinski and Altman 2013) and show all individual data points and express the variation using
 95% confidence interval (95% CI) of the mean. The error bars based on 95% CI of the mean are
 inherently larger than those based on standard error of the mean (SEM) (see figure below for a
 comparison). When the 95% CIs of the mean overlap, data can still be significant at significance
 level of 0.01.

We do agree that the overlapping confidence intervals clearly show the variation within groups, as
 is common with biological material. However, we base our conclusions on measurement from
 multiple tissues from two independent sample sets. A combined analysis of both data sets (see
 figure below) showed that the *mt-Cyb*^{p.D254N} variant causes a biologically and statistically highly
 significant further decrease in CIII activity and maximal OXPHOS capacity in *Bcs1l* mutant mice,
 which is the only theoretically feasible way how it can modify disease progression in the CIII
 deficient mice.

As it seems that we initially failed to present the true scope of CIII activity and respirometry
 measurements in a reader-friendly way, we have now thoroughly revised the manuscript text and
 the figure panels.

References not included in manuscript:

Krzywinski M, Altman N. Points of Significance: Error bars. Nature Methods 2013;10:921–2.

Scatter plots showing values adjusted for sample cohort (F1 P33 2017 and P30 2018 cohort) :
 estimated marginal mean and residuals from two-way ANOVA model (genotype and sample cohort
 as fixed factors). The error bars present 95% CI of the mean.

Reviewer #3 (Remarks to the Author):

The revised paper has resolved some of the questions I raised in my comments. The work,
 especially the genetic results, is convincing in showing a synergistic double-trouble effect of a
 virtually benign polymorphism in cyt b vs. a severe mutation in the UQCRFS1 assembly factor
 *Bcs1l*. As I mentioned in my previous comments, this is an interesting finding although the
 presence of these effects is not an absolutely novelty in the specification of mitochondrial
 phenotypes.

We thank the reviewer for the positive statement about our work, but respectfully disagree about
 his/her view on novelty. This is the only case we are aware of a mitonuclear epistasis affecting the
 same respiratory chain complex - indeed the same subunit - where a mechanism from mouse
 disease progression to molecular level has been delineated. It has major implications for studies of
 mouse disease models in congenic backgrounds, as well as for putative effect of mitochondrial
 haplotypes of human populations on the manifestations of mitochondrial diseases.

The mechanistic explanation offered by the Authors is interesting and supported by some
experimental data obtained in a mtDNA-editable organism such as *R. capsulatus*.

**Reviewer #4 (Remarks to the Author):**

In my opinion the authors significantly improved the manuscript by adding to the revised version
two additional 1 μ s simulations of the protein with the substrates ubiquinone and ubiquinol modeled
at the Q_o and Q_i sites and by analyzing the differences between the monomers.

There are still some minor issues with the paper. The labels in the new figures (S8 and S9) are
misleading. It seems that the authors are referring to different simulations, when in fact the graphs
are for the two monomers. So instead of wild-type 1 and wild-type 2, I would suggest something
like WT monomer 1, WT monomer 2.

We thank the referee for pointing this out. This has now been amended in the revised version.

Furthermore, and since the effects reported in the simulations are subtle, I suggest changing
“Accordingly, we observed a simultaneous reduction” to “a simultaneous small reduction” (Page 8,
line 170)

This is now revised.

Finally, the distance between the dihydroxyquinone in the modeled structures and D254/N254 is
too large for any direct interaction. So the sentence “this is in part due to the vicinity of the QH2
molecule at the Q_o site to the D254/N254 residue (average distance from the simulations between
the Q head group and the CG atom of D254/N254 is $\approx 13.2/11.6$ Å)” should be deleted and instead
the authors should analyze the distance(s) between other ef loop residue(s) or nearby residues that
are closer to QH2. (Page 8, line 177).

We have now revised this section as referee suggested.

**Reviewer #5 (Remarks to the Author):**

The ms thoroughly analysis the effect of a mutant of the bc₁ complex in mice, and propose it is due
to a restricted mention of the Rieske protein, as a result of a D254N mutation.

In spite of the impressive amount of work performed, all data regarding the activity of the bc₁
mutant, including the work performed with a similar mutant in the complex from the bacterium *R.*
*capsulatus*, show only, and this are the authors’ words, subtle effects, if any at all. Therefore, this
raises a key question – is this mutant really important in causing diseases, when the activity of the
complex is barely affected?

It is unfortunate that we failed to present and explain clearly in the manuscript and in the first
response the crucial genetic (crossbreeding) experiment in Fig. 1b and c. See further details in
response to Reviewer 2. Our study exemplifies vividly how a non-pathogenic variant with rather
subtle effect *per se* can have dramatic repercussion in a disease setting.

Also, the molecular dynamics simulations reveal only a very minor effect on the dynamics of the
Rieske protein, when the calculations are performed upon docking the quinone substrates.

Yes, the reviewer is correct in that simulations performed by modeling ubiquinones at Q_o and Q_i
sites reveal lower and subtle differences (Supplementary Fig. 11). However, it is highly likely that
the binding of ubiquinol at Q_o site is also somewhat affected in the first place, as much larger
differences are seen in our simulations without the ubiquinones modeled. Moreover, we have now
discussed additional interactions in the revised text that explains the reason behind reduced mobility

in the case of Qo/Qi sites occupied. Furthermore, and as noted above, the *mt-Cyb*^{p.D254N} is a non-
 pathogenic variant present in a wild-type mouse colony. Therefore, it cannot have a drastic or
 detrimental effect on any parameter on its own.

I am not an expert on Pulsed EPR, but at least the EPR spectra of the WT and mutant complexes are
 also identical.

It is actually quite important that the EPR spectra of the Rieske cluster of the WT and mutant do not
 differ – the unchanged overall shape and position of g transitions in the spectrum reflects the
 integrity of the catalytic Qo site and the proper assembly of the subunits, which is what is expected
 if the complex is to be functional in vivo. Only more sensitive pulsed EPR techniques showed
 subtle differences and these were further discussed, also in the context of their consistency with the
 MD simulations.

Other specific points are:

- Lines 117-118 – If the activity of the bc1 were affected, how the absence of effect on CI-linked
 respiration increased? It should have happened exactly the opposite, if what the authors claim is
 really important

Please, see our response to reviewer 2. In brief, CI is a relatively slow-rate enzyme and insufficient
 to reduce the coenzyme Q pool enough to reveal the CIII deficiency inside respirometry chamber.
 Moreover, NADH cannot be provided directly for CI, as it is impermeable to mitochondrial inner
 membrane, and thus NADH has to be generated indirectly via substrates for TCA cycle enzymes
 which may introduce additional rate-limiting steps into the system. Therefore, a convergent electron
 flow via CI and CII is needed to reveal an acutely sublethal CIII deficiency inside respirometry
 chamber.

- Lines 121-122 – Again, CI-CII linked phosphorylation is not affected by the mutation, neither
 maximal electron transfer capacity, which should have been observed if the D254N mutation would
 be relevant.(lines 141-142).

Please, see our response to the reviewer 2. CI&CII-linked phosphorylation respiration was further
 decreased by *mt-Cyb*^{p.D254N} variant in *Bcs1l*^{p.S78G} mutants as shown by analyses of liver and kidney
 mitochondria from two independent sample sets. The *mt-Cyb*^{p.D254N} variant alone did not decrease
 CI&CII-linked phosphorylation respiration as expected from CIII activity measurements.

- Lines 147-152 – ROS production is also not affected (also in the *R. capsulatus* mutant).

We agree that our data do not suggest increased ROS production by CIII. However, increased ROS
 production from other sources may still play a role.

- In fact, the MD simulations contradict the results – while it is claimed, on its basis, that the
 mobility of the Rieske protein is affected, this does not translate into an effect on its activity!

MD simulations data, especially the simulations performed in the absence of Q/QH₂ molecules (Fig.
 5), showed subtle but clear differences, in line with the experimental data. Additional analysis of
 simulation trajectories revealed that *ef* loop contacts the QH₂ molecule in simulations with Q
 molecules modeled, clearly explaining the cause of reduction in its mobility, with subtle differences
 between wild-type and mutant *ef* loop interactions observed (see revised text). The *Rhodobacter*
 model was employed because it allows investigating subtle structural changes, which were
 predicted by the molecular dynamics simulations, by EPR spectroscopy. The fact that D254N
 (D278N in *Rhodobacter*) did not affect the bacterial bc₁ complex activity in the fully assembled
 enzyme containing stoichiometric amount of RISP is, in fact, completely coherent with the other
 data and subtle nature of the non-pathogenic variant. The D254N variant may not affect a rate-
 limiting step in the catalytical cycle of *Rhodobacter* bc₁ complex, but it clearly does so in the *Bcs1l*

mutant mice with a partial loss of RISP. The partial loss of RISP is expected to lead to CIII
heterodimers with only one active quinol oxidation site. This may render the CIII dimer from *Bcs1l*
mutant mice more sensitive to the effect of D254N variant than WT enzyme. We have now
extended the Discussion related to this.

In summary, all biochemical and respirometry data clearly show that the D254N mutation has
indeed no effect on the mitochondrial respiration of the mice mutants.

We have now made major revisions to the figure panels, as they seem to have been misinterpreted.
The CIII activity measurements showed consistent further decrease in activity in four different
tissues by *mt-Cyb^{p.D254N}* in *Bcs1l* mutant mice. In line with this, respirometry also showed further
decrease in CI&CII-linked phosphorylating respiration. Moreover, the results are consistent across
two independent sample sets. Please, see also response to reviewer 2.

Therefore, some other reasons must exist to explain the phenotypes observed (and those, only in
some tissues).

It is unfortunate that we failed to present and explain clearly in the manuscript and in the first
response the crucial genetic (crossbreeding) experiment in Fig. 1b and c, which shows
unequivocally that the short survival was inherited maternally and therefore must necessarily be
caused by the *mt-Cyb^{p.D254N}* variant. Please, see also response to reviewer 2. We hope that in the
current revised manuscript we present and discuss these crucial data more clearly and thoroughly.

Reviewers' comments:

Reviewer #2 (Remarks to the Author):

In their second revision the authors have provided substantial improvements to the previous revised manuscript, and most of my concerns have been addressed satisfactorily. The authors now present data in a clearer way and show a subtle (though convincing) metabolic phenotype caused by the Cytb variant m.14904G>A alone, which potentially could further aggravate the mitochondrial CIII defects promoted by the p.S78G mutation in the assembly factor Bcs1l.

The main experimental evidence provided in the revised manuscript still does not prove a strong pathogenic effect of the MT-CYTB D254N mutation, as claimed by Reviewer 1 (and myself in my former reviews). This is however compatible with the fact that this mutation in human has been classified as a polymorphism, and as such is only expected to show very (if any) subtle functional effects. The main reason that convinced me was that the authors present very convincing data demonstrating that the short survival trait is maternally inherited. Therefore, it must be caused necessarily by the only difference in the mitochondrial DNA, which is the MT-CYTB D254N mutation. And this mutation by itself seems enough to induce a mild functional defect in complex III, which is clearly amplified in a genetic environment with pathogenic mutations in Bcs1l. To be fair with the authors, maybe I would suggest them to make clear statements emphasizing the maternal inheritance of the short survival trait. Regarding the novelty of the mitonuclear epistasis, the idea is not conceptually new but, to my knowledge, this is the first experimental demonstration in animal models. Therefore, I believe this is an important contribution to the scientific field of mitochondrial biology.

I agree with Reviewer 1 that the respirometry issues remain unsatisfactorily addressed. Respiratory chain activities must be always normalized, as pointed out by reviewer 1, either to citrate synthase activity or to protein concentration, which are the overall accepted normalization ways by both clinical and non-clinical researchers. If the authors find too high inter-group variations, this means that either the whole procedure is technically wrong or that the functional differences claimed by the authors between the experimental groups (control versus mutant mice) are not statistically significant differences, raising concerns about the credibility of the authors' claims. It is just unacceptable to normalize the functional complex III respirometry data by the values obtained for complex IV activities.

Reviewer #5 (Remarks to the Author):

I consider that the authors answered satisfactorily to most reviewers comments, although I still consider that the effects are indeed very subtle and hard to understand that the bacterial complex did not show any major defect. Also, was the assembly of the RISP in the bacterial system analysed ?

A minor point: the supplementary figure legend 12 it should be state dthat the EPR spectrum is for the reduced complexes, ; also, expressions like "ferricyanide spectra" are incorrect: the authors mean spectra of the ferricyanide oxidized sample, etc

Response to reviewers

Reviewer #2 (Remarks to the Author):

In their second revision the authors have provided substantial improvements to the previous revised manuscript, and most of my concerns have been addressed satisfactorily. The authors now present data in a clearer way and show a subtle (though convincing) metabolic phenotype caused by the Cytb variant m.14904G>A alone, which potentially could further aggravate the mitochondrial CIII defects promoted by the p.S78G mutation in the assembly factor Bcs1l.

The main experimental evidence provided in the revised manuscript still does not prove a strong pathogenic effect of the MT-CYTB D254N mutation, as claimed by Reviewer 1 (and myself in my former reviews). This is however compatible with the fact that this mutation in human has been classified as a polymorphism, and as such is only expected to show very (if any) subtle functional effects. The main reason that convinced me was that the authors present very convincing data demonstrating that the short survival trait is maternally inherited. Therefore, it must be caused necessarily by the only difference in the mitochondrial DNA, which is the MT-CYTB D254N mutation. And this mutation by itself seems enough to induce a mild functional defect in complex III, which is clearly amplified in a genetic environment with pathogenic mutations in Bcs1l. To be fair with the authors, maybe I would suggest them to make clear statements emphasizing the maternal inheritance of the short survival trait. Regarding the novelty of the mitonuclear epistasis, the idea is not conceptually new but, to my knowledge, this is the first experimental demonstration in animal models. Therefore, I believe this is an important contribution to the scientific field of mitochondrial biology.

We have now shortened and revised the abstract according to the formatting guidelines of the journal and hope that the current text better highlights the importance of the maternal inheritance.

I agree with Reviewer 1 that the respirometry issues remain unsatisfactorily addressed. Respiratory chain activities must be always normalized, as pointed out by reviewer 1, either to citrate synthase activity or to protein concentration, which are the overall accepted normalization ways by both clinical and non-clinical researchers. If the authors find too high inter-group variations, this means that either the whole procedure is technically wrong or that the functional differences claimed by the authors between the experimental groups (control versus mutant mice) are not statistically significant differences, raising concerns about the credibility of the authors' claims. It is just unacceptable to normalize the functional complex III respirometry data by the values obtained for complex IV activities.

We apologize for not highlighting this sufficiently, but we did provide the key respirometry parameters relative to mitochondrial proteins as a supplement already in the previous revision. We also explained why we find normalization of high-resolution respirometry data to maximal O₂ consumption capacity set by CIV an appropriate and sensitive in-assay normalization. However, we understand the reviewer's wish to see the data in a more traditional way and have revised the manuscript accordingly. We have now added exactly the same respirometry data as in the in the main figures (maximal oxygen consumption capacity set a reference state) also relative to mitochondrial protein (Supplementary Fig. 9).

As for the CIII activity data, we did present the isolated liver and kidney mitochondria data relative to mitochondrial protein throughout all versions of manuscripts. In the case of skeletal muscle and heart homogenates, we chose to normalize CIII activity to CIV activity to minimize technical variation and to be able to quantify small differences reliably. We have found citrate synthase activity to correlate poorly with CIII or CIV activity in hard-to-homogenize tissues (see Figure below, note difference to liver data in lower row). Nevertheless, we remeasured CIII activity from heart and skeletal muscle from the latter mouse panel using a higher number of samples and a more efficient homogenization method (glass-glass homogenizer), and present the data relative to tissue protein (Supplementary Fig. 5). The conclusions remain the same. The related slight text revision are highlighted in red font in the revised manuscript.

Reviewer #5 (Remarks to the Author):

I consider that the authors answered satisfactorily to most reviewers comments, although I still consider that the effects are indeed very subtle and hard to understand that the bacterial complex did not show any major defect. Also, was the assembly of the RISP in the bacterial system analysed?

This is a valid question. Fortunately, the biochemical and spectroscopy data (Supplementary Fig. 12) unequivocally show that RISP was correctly assembled with the other subunits of cytochrome *bc*₁. More specifically,

1. The SDS-PAGE gel on Supplementary Fig. 12a shows all three subunits of the *bc*₁ complex, including RISP. As the protein complex was purified using Strep-tag on cytochrome *b* subunit, the fact that RISP co-purifies with it indicates that it is assembled correctly in the D278N bacterial complex.
2. The EPR spectrum of reduced Rieske cluster of the mutant in membranes shows sensitivity to the occupant of the Q_o site (It has a clear *g*=1.8 transition reminiscent of the interaction of the cluster with oxidized quinone, bound at the Q_o site, and a clear change of the *g* value in response to the presence of the Q_o site-specific inhibitor myxothiazol).
3. The relaxation properties of the cluster assessed by the pulse EPR experiments indicate the influence of the oxidized heme *b*L (enhancement of relaxation). This process strongly depends on distance between heme *b*L and the Rieske cluster, and the fact that it is observed in the mutant indicates that the cluster is in the same distance range as in the native form.

A minor point: the supplementary figure legend 12 it should be state that the EPR spectrum is for the reduced complexes, ; also, expressions like "ferricyanide spectra" are incorrect: the authors mean spectra of the ferricyanide oxidized sample, etc

For clarity, we have correct the figure legend to the following:

Optical difference spectra of hemes b and c of purified wild type cytochrome bc1 complex (WT) and D278N mutant. Red line: ascorbate-reduced minus ferricyanide-oxidized cytochrome bc1; black line: dithionite-reduced minus ferricyanide-oxidized cytochrome bc1.

Comparison of X-band continuous wave EPR spectra of ascorbate-reduced [2Fe–2S] cluster of bc1 complex in chromatophores isolated from wild type (WT) or D278N strain of *R. capsulatus*.

REVIEWERS' COMMENTS:

Reviewer #2 (Remarks to the Author):

In their last revision the authors have provided substantial improvements to the previous versions of this manuscript, and have satisfactorily addressed my concerns. The authors present their data in a clear way and show a subtle (yet convincing) metabolic phenotype caused by the Cytb variant m.14904G>A by itself, which could potentially aggravate the mitochondrial CIII defects promoted by the p.S78G mutation in the assembly factor Bcs1l.

Reviewer #5 (Remarks to the Author):

I keep my main concern - for me, it is not understandable why the mutation on the bacterial complex does not appear to have any significant effect, which strongly suggests that the mutation is, essentially, irrelevant.